# A reconfigurable and conformal liquid sensor for ambulatory cardiac monitoring

Xun Zhao [1,2], Yihao Zhou[1,2], William Kwak[1], Aaron Li[1], Shaolei Wang[1], Marklin Dallenger[1], Songyue Chen[1], Yuqi Zhang[1], Allison Lium[1] & Jun Chen [1] ✉

The severe mismatch between solid bioelectronics and dynamic biological tissues has posed enduring challenges in the biomonitoring community. Here, we developed a reconfigurable liquid cardiac sensor capable of adapting to dynamic biological tissues, facilitating ambulatory cardiac monitoring unhindered by motion artifacts or interference from other biological activities. We employed an ultrahigh-resolution 3D scanning technique to capture tomographic images of the skin on the wrist. Then, we established a theoretical model to gain a deep understanding of the intricate interaction between our reconfigurable sensor and dynamic biological tissues. To properly elucidate the advantages of this sensor, we conducted cardiac monitoring alongside benchmarks such as the electrocardiogram. The liquid cardiac sensor was demonstrated to produce stable signals of high quality (23.1 dB) in ambulatory settings.

Cardiovascular diseases (CVDs) have been the leading cause of premature death, causing tens of millions of deaths each year[1]. Despite its high prevalence, the detection of CVDs has posed significant challenges: frequently, symptoms remain unnoticed or misdiagnosed as significantly less severe conditions, thereby greatly increasing the risk of a cardiac episode[2]. To prevent the onset of the life-threatening effects of CVDs, early monitoring of cardiovascular health through the examination of biomarkers is the first line of defense[3–5]. Traditionally, heart rate and blood pressure are measured through equipment such as cardiac ultrasound[6,7], electrocardiograms[8], sphygmomanometers[9,10], and invasive arterial monitoring[11]; however, these methods are not convenient as they produce only point-in-time measurements and require complicated setups.

Unlike traditional in-office monitoring, ambulatory monitoring allows continuous assessment of cardiac activity over an extended period. Uninterrupted monitoring provides a more comprehensive picture of a patient's cardiac health status, capturing any irregularities or abnormalities that may occur intermittently or during daily activities. Thus, recent wearable bioelectronics such as resistive[12,13], piezoelectric[14,15], magnetoelastic[16], triboelectric[17], capacitive mechanism-based sensors[18–20], and photoplethysmography (PPG) have shown great potential in non-invasive and continuous detection of human

physiological signals such as blood pressure and heart rate[21–23]. However, many of these sensors rely on solid materials for sensing, which tend to be rigid and fail to conform well with the surface of various tissues like skin. This issue is usually circumvented by either applying an external pressure to tightly adhere sensors to the skin surface or by introducing structural and mechanical alterations to match the skin's surface geometry. However, these approaches can result in signal disruptions due to interfacial issues between the solid devices and the dynamic skin surface.

Therefore, there is a significant demand for monitoring techniques that offer continuous, accurate, stable, and accessible cardiac evaluation. Here, we developed an ultrasensitive liquid cardiac sensor for wearable pulse wave monitoring (Fig. 1a). The key to the liquid cardiac sensor is the discovery of permanent fluidic magnets (PFM), a magnetic material that maintains high remanent magnetization and reconfigurability simultaneously[24]. The PFM was achieved by decoupling Brownian motion and stability in magnetic colloidal suspensions, using non-Brownian magnetic particles to construct a three-dimensional (3D) oriented and ramified magnetic (ORM) network structure within a carrier fluid. We systematically investigated the material property criteria, such as temperature, and injection volume, for PFM to conform to the hierarchical structures on the human skin.

[1]Department of Bioengineering, University of California, Los Angeles, Los Angeles, CA, USA. [2]These authors contributed equally: Xun Zhao, Yihao Zhou.
✉ e-mail: jun.chen@ucla.edu

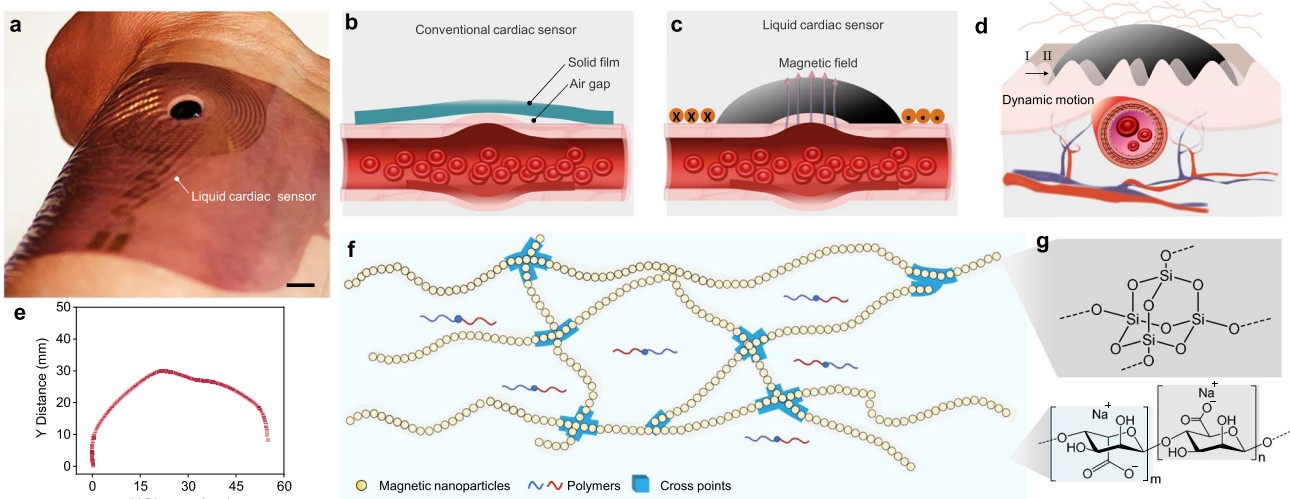

**Fig. 1 | A reconfigurable and conformal liquid cardiac sensor. a** Picture of the liquid cardiac sensor. Scale bar, 1 cm. **b** Schematic showing the gap between conventional solid bioelectronics and skin surface. **c** Schematic showing the reconfigurability of the liquid cardiac sensor droplet on a topographically complex surface. **d** Schematic demonstrating excellent conformation of the liquid cardiac sensor to the dynamic skin surface. **e** Plots of the outline of the skin surface. **f** Diagram showing the alignment of magnetic nanoparticles into a chain network within the PFM. **g** Chemical structure diagram of carrier fluids: silicon dioxide, and sodium alginate. Source data are provided as a Source Data file.

These findings lay a firm foundation for the development of PFM-based soft bioelectronics. We identified that PFM, made from alginate solution, can form a seamless interface to the wrinkles and creases on the skin surface. When applied to the wrist for pulse wave measurement with a conformal soft coil, the PFM liquid bioelectronics yields a stable electrical output even during biomechanical movement, enabling highly sensitive and precise monitoring of cardiovascular systems with mitigated motion artifacts. This superior performance of PFM-based soft bioelectronics is attributed to the unique integration of flowability and magnetic functionality inherent to the PFM material. On one hand, PFM is flowable, allowing it to fill the hierarchical creases and wrinkles, and conform to the complex geometry of skin surfaces. On the other hand, despite undergoing shape changes during the interface filling process, the magnetization of PFM remains intact, enabling the sensing functionality. In addition, PFM can readapt to the newly generated wrinkles and grooves and, therefore, accommodate the skin's dynamic surfaces during biomechanical movement benefiting from its reconfigurability. Overall, the PFM-based reconfigurable and conformal liquid sensor has the potential to largely improve the diagnosis of CVDs and other physiological conditions. With the material-to-sensor strategy and the established criteria for achieving a seamless bioelectronic-tissue interface, this work is a milestone in designing soft and conformal bioelectronics, facilitating the transformation of current healthcare based on disease management to a more personalized mode.

## Results

### Creating a liquid cardiac sensor

The skin's surface is characterized by a complex topography with various contours, curves, and irregularities, such as wrinkles, grooves, and creases, which are present at rest and become more pronounced during biomechanical movement. Conventional solid materials fail to adequately adapt their geometry to match these skin contours, which could result in noticeable air gaps, leading to noise, low-accuracy measurement, and even electrical spikes (Fig. 1b). The liquid PFM is reconfigurable and can easily adapt to the skin contours automatically without the need of external pressure (Fig. 1c). This reconfigurability allows them to conform well to the wrinkled skin surface for accurate measurement (Fig. 1d and Supplementary Fig. 1). In

experiments, the curvature of the wrist is visible in the 3D scanning data (Fig. 1e).

Creating liquid bioelectronics capable of generating continuously variable magnetic flux relies on the concept of an oriented and ramified magnetic (ORM) network structure dispersed throughout the PFM (Fig. 1f), in which magnetic nanoparticles are aligned along the chain directions. Biocompatibility of the bioelectronics can be enhanced by coating the magnetic nanoparticles with a protective layer whose chemical structure and formula are illustrated in Fig. 1g. A 3D illustration of the chain structure and the orientation of its constituent magnetic nanoparticles are further displayed in Supplementary Fig. 2. The self-assembled 3D ORM network structure plays three critical roles in creating the PFM with both material stability and permanent magnetization. First, it mitigates the gravity-settling effect of individual nanoparticles. Second, it conveys structural and mechanical support to maintain the stability of the colloidal system. Third, it allows macroscopic permanent magnetization to emerge from the microstructural orientation of the magnetic nanoparticles. To characterize the morphology of the created PFM, microscopy images of the 3D ORM network structure were taken under external magnetic fields (Supplementary Fig. 3).

We further investigated the 3D ORM network structure formations under a hysteresis loop at different temperature variations. First, magnetic nanoparticles were coated with a layer of silica (Supplementary Fig. 4), which were originally evenly arranged in the carrier fluids before magnetization. Then, it underwent a gradual controlled cooling process from approximately 47 °C to 13 °C (Fig. 2a). The impact of temperature changes on the magnetic property can be observed in Fig. 2b, c. Based on the hysteresis loop of PFM, it shows slightly higher coercivity as the temperature decreases. The outlined region in Fig. 2b was magnified in Fig. 2c, offering a clear view of the variation in coercivity across different temperatures. Furthermore, the influence of an impulse magnetic field determines the effect of z-axis magnetization on the 3D ORM structure (Fig. 2d). This was achieved by applying an electrical pulse through a capacitor to produce a magnetic pulse. A syringe filled with PFM was placed inside a coil for material manipulation, as depicted in the lower portion of Fig. 2e.

Droplets with various magnetic field distributions can be achieved by positioning them at different angles. Consequently, a finite element

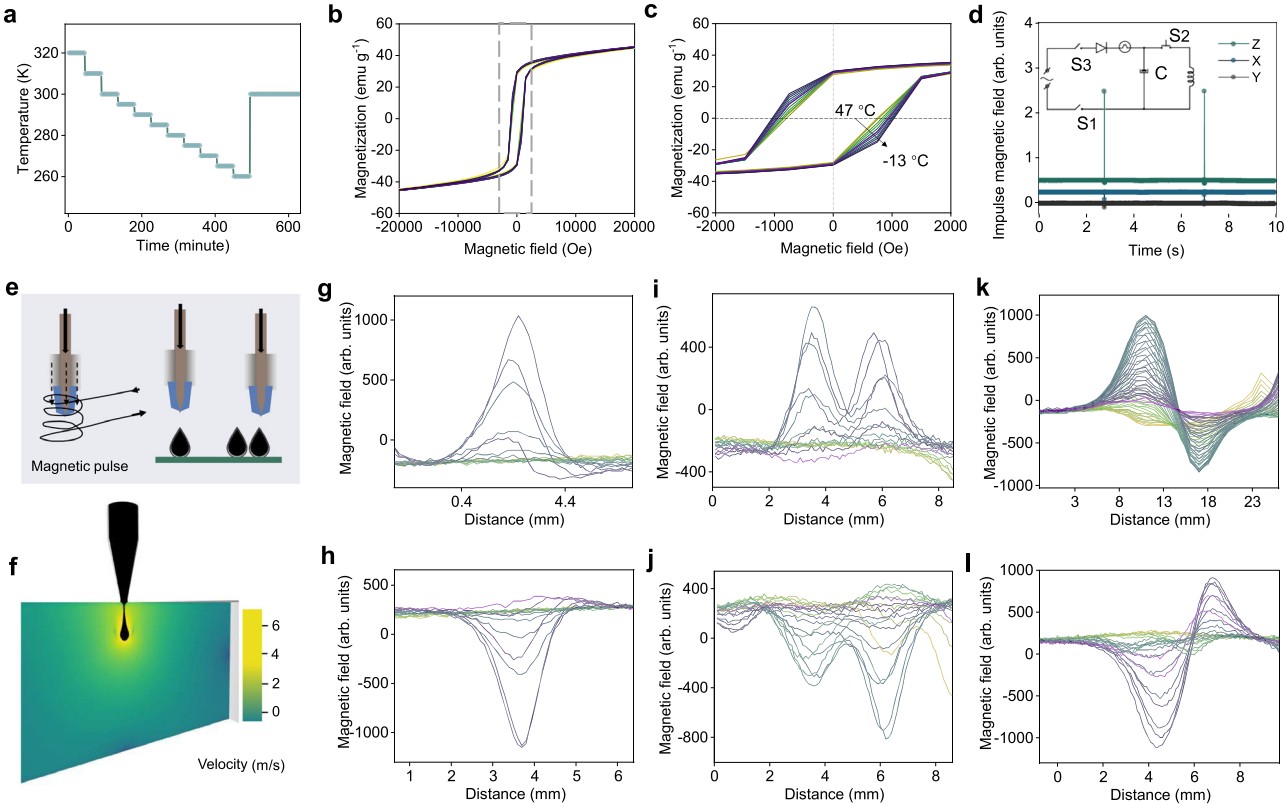

**Fig. 2 | Dynamic magnetic field and temperature influence on the 3D ORM network structure. a** ORM network structure at different temperatures ranging from 320 K (≈ 47 °C) to 260 K (≈ − 13 °C). **b** Hysteresis loop at different temperatures. **c** Influence of temperature change from 320 K (≈ 47 °C) to 260 K (≈ − 13 °C) in the magnified area of the gray square. **d** Magnetic pulse generated specifically in the z-direction to magnetize the ORM network structure. The inset figure is the diagram of the magnetic pulse generation by the capacitor. **e** Diagram of a syringe filled with PFM placing inside a coil for material manipulation. **f** Finite element analysis was constructed to simulate the injection process. **g** Magnetic field of the single droplet. **h** Magnetic field of the single droplet magnetized in a different direction. **i** Magnetic field of two droplets. **j** Magnetic field of two droplets magnetized in another direction. **k** Magnetic field of the tilted droplet. **l** Magnetic field of the tilted droplet magnetized in another direction. Source data are provided as a Source Data file.

analysis was constructed to simulate this process, incorporating parameters such as velocity (Fig. 2f and Supplementary Fig. 5) and droplet volume (Supplementary Fig. 6). The results indicate that the droplet maintains its shape throughout the injection process. Experimentally, we measured the magnetic field of a droplet of PFM. The magnetic field of the droplet was elucidated in Fig. 2g. The bulk of the field was concentrated around the center of the droplet where there was the highest volume of magnetic nanoparticles, showing that larger volumes of PFM could be deposited to induce greater magnetic flux. Figure 2h illustrates the magnetic field when the droplet was magnetized in a different orientation. Figure 2i shows the magnetic field of two contiguous droplets. Figure 2j displayed the magnetic field when the two contiguous droplets were magnetized in an opposite orientation. Finally, Fig. 2k, l showed the magnetic field of a tilted droplet and its different magnetization states. These diverse patterns demonstrate the versatility of the PFM. The long-term durability of the magnetic field density of the PFM was also assessed, revealing a minimal decrease over a testing period exceeding 80 days (Supplementary Fig. 7).

**Understanding of the magnetic property of the PFM**

We investigated the magnetic behavior of a PFM droplet in three dimensions by observing the micro-computed tomography image. Figure 3a demonstrates the microscope image of the 3D ORM network structure. The corresponding micro-computed tomography image is shown in Fig. 3b–d. We also measured the produced magnetic field of the droplet in three dimensions. Through characterizing the magnetic field of the PFM, we could better understand magnetic flux changes induced by pulse waves in patients.

The PFM consisted of a 3D ORM network structure, which allowed it to maintain ferromagnetism and stability simultaneously. To better understand the magnetic field produced by the 3D ORM network, magnetic fields in all three dimensions were measured (x-, y-, and z-directions, Supplementary Fig. 8). Using a droplet of PFM, we measured the produced magnetic field repeatedly (Supplementary Fig. 9). In the x-direction (Supplementary Fig. 9a), the PFM displayed a positive magnetic field for the first 5 mm and a negative magnetic field for the last 5 mm, both with the highest peak magnitudes of around 800 arb. units. The results indicated that the magnetic field of the PFM droplet was evenly dispersed in the x-direction. In contrast, the y-direction (Supplementary Fig. 9b) exhibited a peak magnetic field of around 800 arb. units in magnitude for the first 5 mm measurement but showed alternating polarity. The observed magnetic field output in the z-direction was more constant with a positive peak of around 900 arb. units at the 5 mm measurement (Supplementary Fig. 9c). As indicated by the figures, the PFM droplet exhibited variation in magnetic field along different axes. Due to the variation, the spatial changes in the magnetic field could be easily mapped, allowing for a unique representation of information based on the direction of change. They also aid in identifying the magnetic field during the resting period before a pulse, which is crucial for accurate waveform measurements.

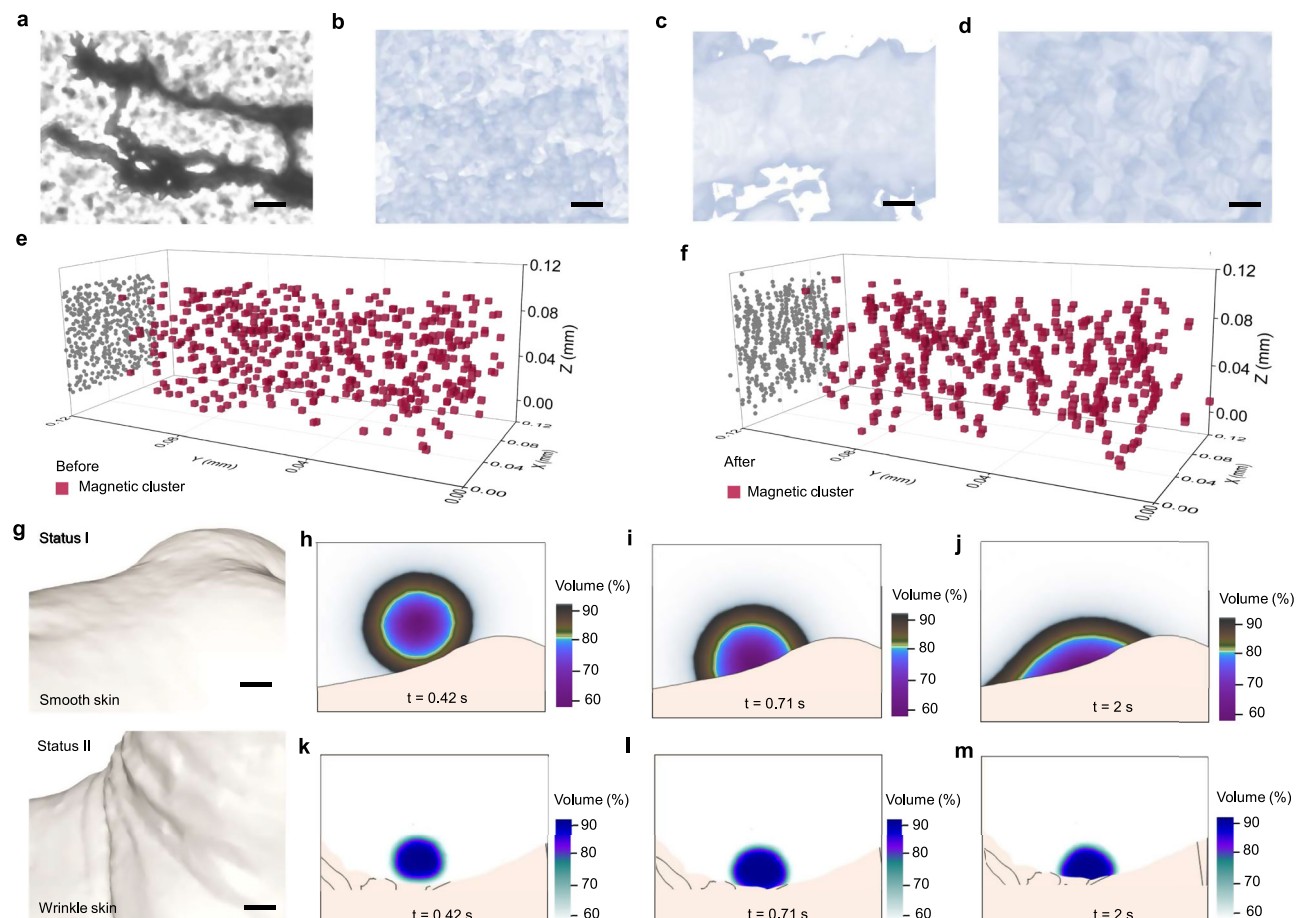

**Fig. 3 | Magnetic field characterization of the PFM droplets. a** Microscope image of the 3D ORM network structure. Scale bar, 10 μm. **b–d** Micro-computed tomography image of the network structure. Scale bar, 5 μm. All images were conducted following at least three independent experiments. Those images yield similar results in different independent experiments. Typical images are shown in the figure. **e, f** Monte Carlo simulation for the formation of the network structure. **g** Tomographic images of the skin on the wrist in two different statuses. Scale bars, 4 mm. **h–j** Finite element simulation of the liquid cardiac sensor applied to the wrist. **k–m** Finite element simulation of the liquid cardiac sensor is applied to the wrist in status II. Source data are provided as a Source Data file.

We also utilized Monte Carlo simulation to replicate the formation of the 3D ORM network structure (Supplementary Note 1). The initial state is illustrated in Fig. 3e, depicting particles randomly dispersed in the space. Following the simulation, Fig. 3f reveals the emergence of chain structures. The illustrated comparison of the projected chain structures was illustrated (Supplementary Fig. 10). To observe the behavior of multiple PFM droplets arranged together, point-in-time measurements of 12 PFM droplets arranged in a 3 by 4 grid pattern were obtained in the x-, y-, and z-directions, respectively (Supplementary Fig. 11). Opposing magnetic fields with equal magnitude were observed between the PFM droplets, indicated by the dark blue areas in the figures. The spatial mappings of magnetic field revealed that the magnetic field of individual droplets could contribute to the field detected on neighboring droplets. In pulse wave monitoring, multiple droplets could be used to obtain a more comprehensive measurement of the cardiovascular system.

To investigate how the liquid cardiac sensor fills the gaps in the skin surface, we employed an ultrahigh-resolution 3D scanning technique to capture tomographic images of the skin on the wrist (Fig. 3g). Then, we established a theoretical model based on Navier-Stokes equations to gain a deep understanding of the intricate interaction between our reconfigurable sensor and dynamic biological tissue (Supplementary Note 2). During biomechanical movement, the skin will develop wrinkles, potentially causing the device to separate from its surface. Gaps were present between solid devices and skin due to

the absence of an external pressure applied to the solid devices (Supplementary Fig. 12a). However, images demonstrated that the PFM could readily fill in the skin's grooves (Supplementary Fig. 12b).

We utilized the finite element simulation to investigate the contact between these two different states. Initially, as the liquid cardiac sensor is applied to the wrist, it progressively fills the grooves (Fig. 3h–j). Subsequently, as the wrist moves and wrinkles form, the surface tension of the liquid ensures intimate contact with the skin surface (Fig. 3k–m). We also calculated how the liquid cardiac sensor was released on the wrist (Supplementary Fig. 13). The reproducibility of the shape of the liquid sensor was investigated during each injection (Supplementary Note 3). Despite variations in different places, the final formed shapes were similar to each other.

Taking a further step, we also investigated how the magnetic field changes in response to external stretching. To demonstrate this, we measured the magnetic field distribution when the sensor was stretched on the skin during motion. In our experiments, we placed the liquid sensor on the artificial skin and stretched it by ~10%. The magnetic mapping of the liquid sensor was measured before and after stretching. First, we measured the magnetic mapping of the liquid sensor along its z-axis (Fig. 4a, b). The results showed a decrease in the sensor's magnetic field and an increase in the profile of the droplet. We also measured the magnetic field strength (Fig. 4c), which decreased from 6.05 mT to 5.54 mT, accounting for 91.5% of its initial magnetic

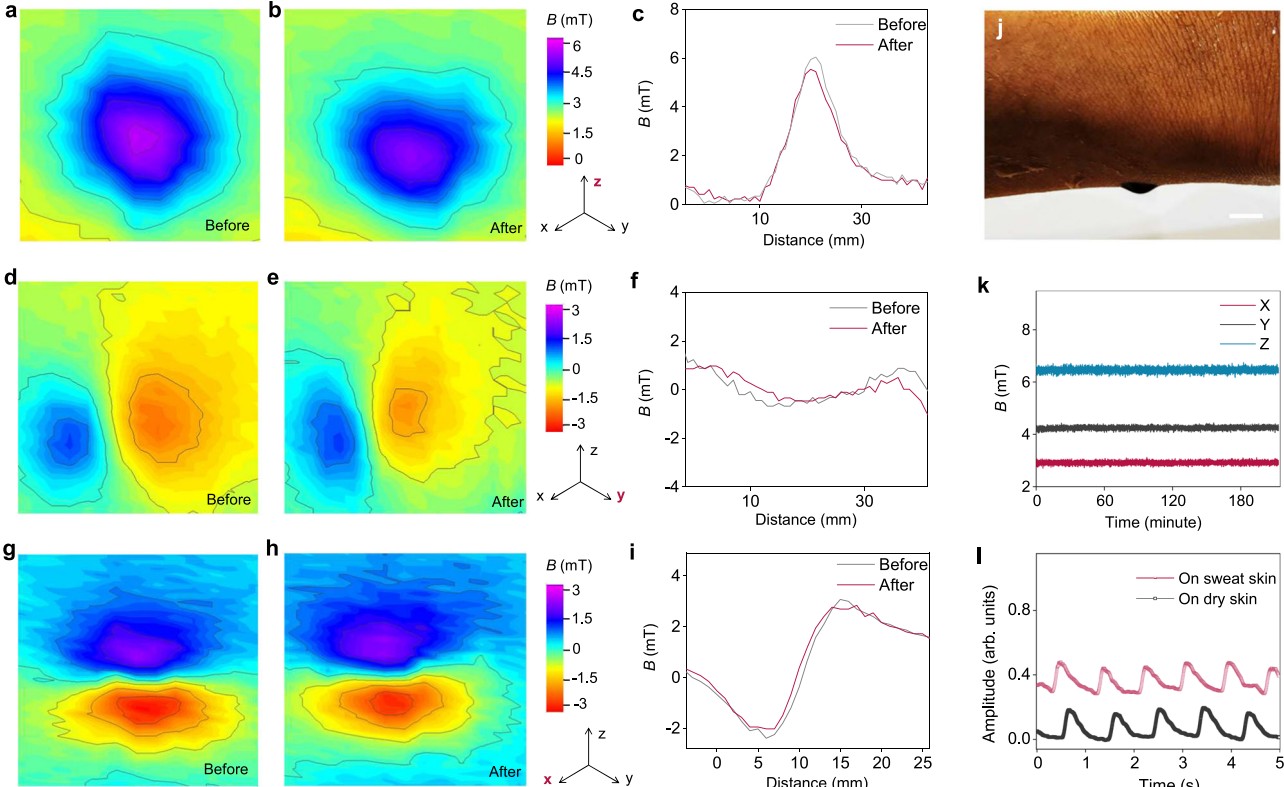

**Fig. 4 | Magnetic mapping of the liquid cardiac sensor.** Magnetic mapping of the liquid sensor along its *z*-axis before (**a**) and after stretching (**b**). **c** Magnetic field strength of the liquid sensor along its *z*-axis before and after stretching. Magnetic mapping of the liquid sensor along its *y*-axis before (**d**) and after stretching (**e**). **f** Magnetic field strength of the liquid sensor along its *y*-axis before and after stretching. Magnetic mapping of the liquid sensor along its *x*-axis before (**g**) and

after stretching (**h**). **i** Magnetic field strength of the liquid sensor along its *x*-axis before and after stretching. **j** Liquid sensor was flipped over when it was attached to wet skin. Scale bar, 5 mm. **k** Magnetic field of the PFM measured for three hours. **l** Liquid cardiac sensors tested on dry skin and on sweaty skin. Source data are provided as a Source Data file.

field. The magnetic field mapping along the *y*-axis was also measured (Fig. 4d, e), showing a similar decrease (Fig. 4f). In the *x*-axis, the decrease was less compared to the other two directions (Fig. 4g–i). This might be due to the stretching being applied on the *y*-axis. As a result, stretching the liquid sensor will slightly decrease its magnetic field, but overall, it will not compromise its performance as it can still maintain 91.5% of its initial magnetic field and we use the magnetic field variation for sensing.

Liquid cardiac sensors were applied on both dry and wet skin to perform comparative tests to examine their robustness in both scenarios. The sensor can maintain firm contact with the wet skin even when flipped over (Fig. 4j and Supplementary Fig. 14). After spraying with artificial sweat, we measured its magnetic field during the drying process. Initially, we observed slight fluctuations, but the magnetic field stabilized within the following three hours, demonstrating that the liquid sensor can operate continuously against sweat (Fig. 4k). In addition, we also measured the generated signals on both dry and wet skin, and in both scenarios, the liquid sensor delivered stable measurement signals (Fig. 4l). As a result, the presence of sweat did not show a negative impact on the test accuracy.

**Wearable cardiac monitoring**
Human pulse waves result from the rhythmic expansion and contraction of arteries through the passage of blood and can be readily detected at various points throughout the body. Biomarkers obtained from human pulse waves provide crucial insight into hemodynamic parameters such as cardiovascular pulse waves and stroke volume. Liquid bioelectronics utilizing PFMs can capture subtle biomechanical signals such as pulse waves through measuring magnetic flux

variations, which can be detected by conformal soft coils to generate resolvable electrical signals. A primary advantage of liquid-based bioelectronics is their near-perfect conformability to complex surfaces such as human skin (Fig. 5a, b). While recent developments to wearable devices have greatly enhanced conformability, the existence of a gap at the interface of human skin and device diminishes device sensitivity (Supplementary Fig. 15). Potential solutions such as applying pressure to the top of the device or increasing binding strength may enhance signal resolution, but this presents disadvantages such as worsened wearing comfort.

To obtain the electrical signals from the PFM, we used a conformal coil attached to the skin to detect the magnetic field variations from the PFM (Fig. 5c and Supplementary Video 1). Then, in order to measure the subtle pulse waves, a biosensing board was applied to connect to the conformal coil (Supplementary Fig. 16). The diagram of the biosensing board was illustrated in Fig. 5d and Supplementary Fig. 17. This board can continuously work for ~ 55.31 h after being fully charged once (Supplementary Note 4). A smart device such as a laptop connected to the biosensing board is used to record the signals for further pre-processing and to detect the presence of abnormalities in the pulse wave profile (Supplementary Note 5). To rule out the interference from ambient magnetic field, we also measured the pulse wave with only the soft coil. It did not yield any electrical signals (Fig. 5e). Thus, the soft coil satisfies our study requirements as its resistance remains almost unaltered to pulse waves, thereby minimizing possible interference.

To assess the functionality of the liquid cardiac sensor, it was evaluated alongside an electrocardiogram (ECG): the benchmark medical testing device used to measure electrical heart activity over

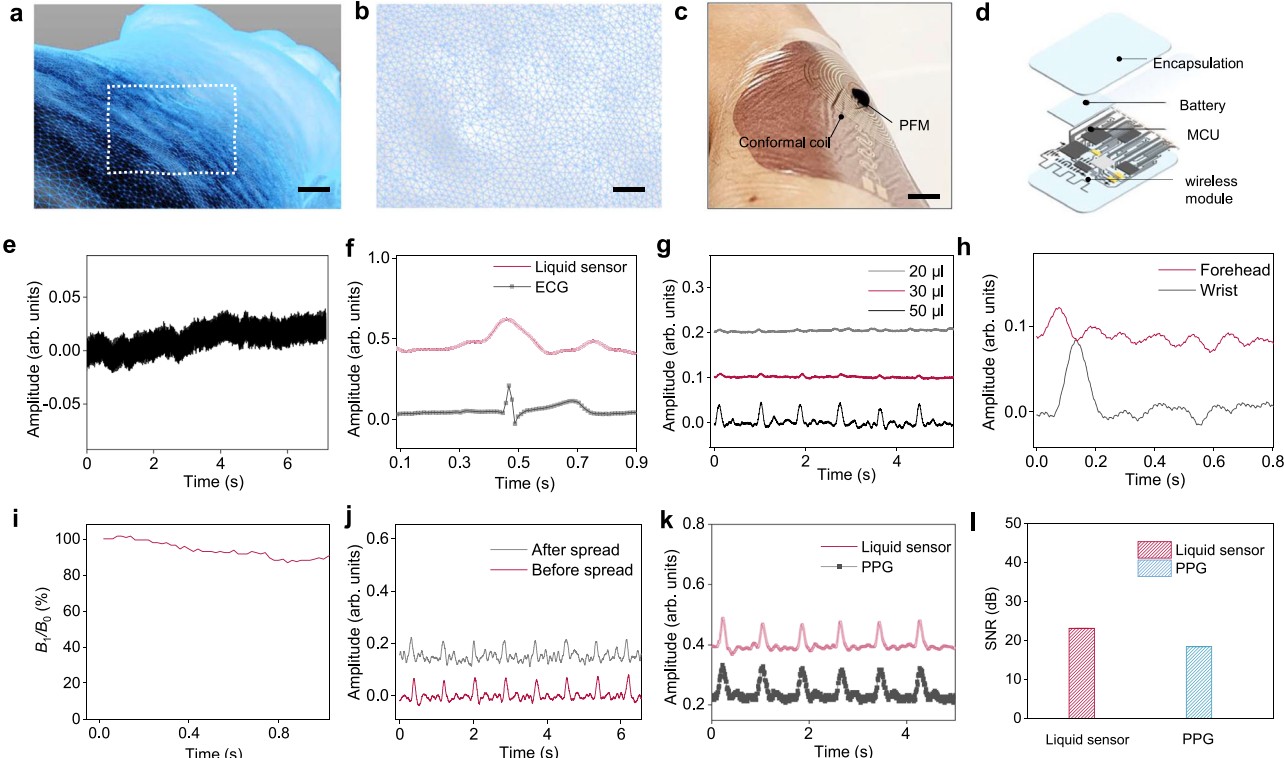

**Fig. 5 | The PFM based liquid sensor for continuous cardiac monitoring. a** 3D scanning technique to capture tomographic images of the skin surface. Scale bar, 8 mm. **b** Enlarged view of the skin surface. Scale bar, 3 mm. **c** Image of the liquid cardiac sensor on the wrist. Scale bar, 8 mm. **d** Circuit diagram of liquid cardiac sensor, including microcontroller (MCU), encapsulation, battery, and wireless module. **e** Using only the soft coil for pulse wave monitoring. **f** Characteristics of pulse wave from the liquid sensor. **g** Testing of the liquid sensor with different PFM volumes of 20, 30, and 50 μl. **h** Liquid sensors measured simultaneously at two sites: one on the forehead and one on the wrist. **i** Magnetic field of the PFM was measured when it was slightly spread. **j** Liquid sensor tested before spread and after spread. **k** Comparison between the liquid sensor and a gold-standard photoplethysmography (PPG) sensor. **l** Comparison of our device and PPG sensors with respect to their SNR. Source data are provided as a Source Data file.

time (Supplementary Fig. 18). The liquid cardiac sensor provides stable signals and can be combined with the ECG measurement to provide a comprehensive evaluation of the cardiovascular health status of human beings. As demonstrated in Fig. 5f, the liquid cardiac sensor accurately captured the pulse wave at the same frequency as the ECG. The ECG records the electrical signals of the heart beating, while the PFM liquid sensor records the pulse wave containing information of changes in blood flow caused by the heart pumping. We have compared the quality of the recordings with PFM volumes of 20, 30, and 50 μl. With the increase in volume, the liquid cardiac sensor demonstrates higher electrical output and better signal quality (Fig. 5g). This is because as the volume of PFM increases, it forms a stronger magnetic field, thereby generating higher electrical output.

To show its broad application, we also conducted a two-site measurement simultaneously, with one site on the forehead and another on the wrist. By measuring the pulse at these two sites with two droplets of the PFM, we observed a clear time lapse of 0.06 sec and amplitude differences between the two signals in the two locations (Fig. 5h). This is due to the different distances from the ventricular ejection site to the two measurement sites. In addition, the pulse wave is weaker in the head compared to the wrist, possibly due to vascular differences in pressure. This finding proves that the multiple PFM liquid sensors can form a body area sensor network for blood pressure, pulse wave velocity, arterial wall stiffness measurements, among others. To understand the performance of the liquid cardiac sensor after it was spread, we assessed its magnetic field strength. The experiment demonstrated that the magnetic field strength decreased

when the liquid was spread (Fig. 5i). It also resulted in reduced sensing performance, leading to a slight decrease in recording quality but within an acceptable level (Fig. 5j).

In comparison to the gold standard in clinical settings such as the PPG pulse wave monitoring device, both our device and the PPG generated stable signals and matched well during testing (Fig. 5k and Supplementary Fig. 19), indicating that the liquid cardiac sensor performed well. The liquid sensor was tested to have an SNR of 23.1 which is slightly higher than the PPG's SNR of 18.4 (Fig. 5l). It is important to highlight that no external pressure is required between the liquid cardiac sensor and the skin for measurement. In contrast, solid capacitive-based sensors necessitate external pressure to obtain the pulse wave signals (Supplementary Fig. 20).

After the measurement, we can easily wipe out the liquid sensor from human skin. To demonstrate this, we tested the liquid cardiac sensor on an ex vivo swine model. First, we applied the PFM to porcine skin (Supplementary Fig. 21a, b). Then, it was rinsed with water (Supplementary Fig. 21c). The corresponding microscope images are shown in Supplementary Fig. 21d–f. There were no residues on the skin, proving that the PFM is easy to be removed after use. In addition, we used a video to illustrate how to remove the device after the measurement (Supplementary Video 2).

## Discussion
Existing wearable sensors rely on solid materials that are rigid and cannot adapt to the dynamic epidermal surface, limiting their capabilities for pulse wave monitoring and cardiovascular diseases screening. The PFM will revolutionize materials-driven

technology and can be applied to various liquid state devices. In addition, the liquid cardiac sensor will offer crucial insights into hemodynamic parameters such as cardiovascular pulse waves and stroke volume. Simultaneously, the rapid advancement of artificial intelligence technologies and machine learning algorithms has greatly enhanced the accuracy of wearable sensors, facilitating improved disease and risk management in patients. With more stable cardiovascular signals, patients can be measured with exceptional precision, enabling real-time diagnosis of CVDs and providing continuous feedback for timely intervention.

PFMs offer a promising approach to solving interfacial issues and thus enhance the ability to measure the pulse waves. Showing comparable reliability and sensitivity to industry standards, such as the PPG, the PFM based liquid cardiac sensor recorded pulse waves with minimal interference and even in an ambulatory state, renovating the current solid bioelectronics for pulse wave monitoring. Overall, the strategies employed in the design of the PFM can hopefully be applied to future platform technologies and ameliorate monitoring obstacles for the delivery of high-quality healthcare.

## Methods

### PFM synthesis

The neodymium-iron-boron (NdFeB) magnetic nanoparticles, with a diameter of 100 nm, were purchased from Nanoshel Inc. In order to coat the NdFeB with a layer of silicon dioxide, NdFeB was mixed with tetraethyl orthosilicate (TEOS, Sigma Aldrich) under continuous stirring during the hydrolysis and polycondensation reaction. Subsequently, the NdFeB magnetic nanoparticles were combined with 2-5 wt.% sodium alginate (Sigma Aldrich) as the carrier fluid. The volume percentage of the NdFeB ranged from 1 vol.% to 28 vol.%. After thorough mixing for 10 min, the mixture was uniformly dispersed using an ultrasonic homogenizer (FS-550T) operating at a power of 550 W for three hours. Following this step, magnetization was performed on the mixture using impulse fields ranging from 0.01 T to 3 T by means of an impulse magnetizer (IM-10-30, ASC Scientific).

### Magnetic property characterization

A custom-made platform was designed for precise observation of the PFM. Magnetic hysteresis loop measurements were conducted using a superconducting quantum interference device (SQUID, Quantum Design, MPMS3) magnetometer. The magnetic flux density map was measured with a stage equipped with a hall sensor array (MLX 90393, Melexis), providing an improved resolution of $0.161 \mu T$. Transmission electron microscopy (FEI, TF20 High-Resolution EMT) was employed to observe the morphology of the magnetic nanoparticles.

### Fabrication of the liquid cardiac sensor

A gold target was sputtered onto both sides of the soft substrate, which was then placed on a flat surface. Subsequently, it was marked by the fiber laser (FMM-3RW2-U1, OMTech Laser) to remove residues from the substrate's two sides, leaving behind a helix coil. Then, it was connected with copper wire by conductive silver ink (Conduction Inc.). The pattern is depicted in Supplementary Fig. 22. The liquid was directly put on bare skin as there was a hole in the polyurethane film manufactured with a laser.

### 3D scanning to capture the curvature of the wrist

The maximum scanning depth was set to be 0.3 m, with a resolution of 2048 by 2048 pixels. The scanning device has a sensor size of 35.8 mm and a focal length of 30 mm. The scanning speed was set at 2 Hz, and the entire scanning process took 2 minutes for a single scan. Afterward, all the raw data was processed using Adobe Substance 3D Sampler tools for model reconstruction sequentially.

### Artificial perspiration

Artificial perspiration was used to test the stability of the liquid cardiac sensor. The preparation involved adding 4.65 g NaCl (Sigma Aldrich), 3.87 g 1 M lactic acid solution (Alfa Aesar), 1.80 g urea (Alfa Aesar), 1.37 g KCl (Sigma Aldrich), 0.756 g $NaHCO_3$ (Sigma Aldrich), 0.546 g 1 M $NH_3 \cdot H_2O$ (Sigma Aldrich), 0.175 g $Na_2SO_4$ (Sigma Aldrich), and 0.0276 g uric acid (Alfa Aesar) to 3 L of deionized water and mixing for 30 min.

### Materials characterization

The morphology of the magnetic nanoparticles were characterized by scanning electron microscopy (Supra 40VP, Zeiss). The morphology of the PFM was characterized by a Zeiss inverted microscope (Zeiss Axio Observer Z1) and in-house micro-computed tomography.

### Electrical output measurement

The voltage and current signals of the liquid cardiac sensor were measured by a voltage preamplifier (SR560, Stanford) and current preamplifier (SR570, Stanford), respectively. All the data acquisition boards were connected to one laptop, and a Python program was used to record all the data simultaneously. The data were synchronized using a common time axis. Our liquid cardiac sensor was compared with a gold-standard PPG pulse wave monitoring device equipped with a Surface-Mount Ambient Light Photo Sensor (APDS-9008).

### Circuitry design

For the integrated signal recording circuit, first, the electrical signals from the liquid sensor patch were amplified using an analog circuit (AD620). A voltage shifter (LM358) was connected to the amplifier to shift the voltage from 0 V to 2.5 V, and a voltage follower (LM358) was used to stabilize the voltage. Subsequently, the analog signals were converted to digital signals by the embedded 12-bit analog-to-digital converter in the microcontroller (SAMD21G18) and then transmitted to the receiver unit through a 2.4 GHz wireless module (NRF24L01). The data was recorded on a laptop using a Python program. Finally, the data was processed by MATLAB 2022b for signal analysis. The biosensing board was attached to the skin using a polyurethane film (MarvellHealth) with a size of 5 cm by 4 cm.

### Test on human subjects

The PFM droplet was encapsulated into a syringe to be placed onto the wrist. A conformal coil with a diameter of 20 mm was printed onto a medical thin film and was finally laminated on the skin's epidermis. All cardiovascular monitoring was performed using human subjects in compliance with all the ethical regulations under a protocol (ID: 20-001882) that was approved by the Institutional Review Board (IRB) at the University of California, Los Angeles. The informed consents were obtained from all participants in this study. All participating subjects in the study belonged to the University of California, Los Angeles with the age from 20–30 years old. Population characteristics are not relevant to the experiment or the results of this study.

### Statistics and reproducibility

No statistical methods were used to pre-determine sample sizes, but our sample sizes are similar to those reported in previous publications[25]. The data met the assumptions of the statistical tests used, and the normality and equal variances were formally tested. All studies were conducted following at least three independent experiments, including the scanning electron microscope image, transmission electron microscopy image, microscope image, immunofluorescence staining image, and fluorescence microscope image. Those images yield similar results in different independent experiments. Typical images were shown in the figures.

**Reporting summary**

Further information on research design is available in the Nature Portfolio Reporting Summary linked to this article.

## Data availability

All data supporting the findings of this study are available within the article and its supplementary files. Any additional requests for information can be directed to and will be fulfilled by, the corresponding authors. Source data are provided in this paper.

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

## Acknowledgements

J.C. acknowledges the Henry Samueli School of Engineering & Applied Science and the Department of Bioengineering at the University of California, Los Angeles, for their startup support. J.C. acknowledges the Vernroy Makoto Watanabe Excellence in Research Award at the UCLA Samueli School of Engineering, the Office of Naval Research Young Investigator Award (Award ID: N00014-24-1-2065), National Institutes of Health Grant (Award ID: R01 CA287326), National Science Foundation Grant (Award Number: 2425858), the American Heart Association Innovative Project Award (Award ID: 23IPA1054908), the American Heart Association Transformational Project Award (Award ID: 23TPA1141360), the American Heart Association's Second Century Early Faculty Independence Award (Award ID: 23SCEFIA1157587), the Brain & Behavior Research Foundation Young Investigator Grant (Grant Number: 30944), and the NIH National Center for Advancing Translational Science UCLA CTSI (Grant Number: KL2TR001882).

## Author contributions

J.C. guided the whole research project. X. Z., Yihao Z., and J.C. conceived the idea, designed the experiment, analyzed the data, drew the figures, and wrote the manuscript. W. K., Aaron L., M. D., S. C., Yuqi. Z., S. W., and Allison L. assisted in device fabrication and testing. All authors have read the paper, agreed to its content, and approved the submission.

## Competing interests

A US patent 63/596,815 has been filed related to this work from the University of California, Los Angeles. The remaining authors declare no competing interests.
