## [Peer Review File · Nature Communications]

REVIEWER COMMENTS

Reviewer #1 (Remarks to the Author):

The authors presented a liquid sensor for cardiac monitoring to adapt with the interfaces of biological tissues. This clever design is supported by several proof-of-concept experiments. Overall this is a well-organized and well-written manuscript with a compelling idea and well-designed supporting figures. However, there were a few major points of concern that I suggest are corrected before this paper is accepted for publication.

1. The authors claimed that the cardiac sensor can withstand against interference such as respiration. However, it is well known that high-resolution cardiac waveforms naturally contain information of respiration, and several techniques have been developed to measure respiration rate from cardiac waveforms. It is therefore significant to check the results in this study, and modify the claims in both abstract and introduction accordingly.
2. The monitoring of the cardiac sensor is achieved through magnetic field, but when the patch is stretched and twisted with skin during motion, the magnetic field generated will drift accordingly. It is recommended to add dynamic magnetic field distribution measurements during motion.
3. In Fig 3h-j, the authors conducted finite element simulation of the liquid when applied to the wrist. How long does it take to evolve from h to j? In addition, what is the experimental surface tension of the liquid? When a liquid is dispersed onto skin due to surface tension, it is natural for the liquid to disperse differently each time during placement. Have the authors examined the reproducibility of the shape of such liquid sensor both in simulation and experiments?
4. When the subject experiences perspiration, will sweat go into the liquid cardiac sensor and influence the measurement accuracy? In addition, the humidity of skin when placing the sensor matters – it is recommended to conduct comparative experiments on both dry and wet skin and examine the measuring accuracy.
5. What is the sampling frequency of the liquid sensor shown in Fig 4? Will the power consumption and Bluetooth transmission become a challenge when measuring such waveform continuously? It is suggested to add the lifetime of continuous measurements in the main text.
6. In supplementary fig 2, it is suggested to add labels for atoms and the dimensions of the structure.
7. In supplementary fig 4, it is suggested to add the magnified area label of b from a. Upon comparing a (which are two spheres) and the zoomed b (which looks like irregular squares), the shape and morphology don't match each other.
8. In supplementary fig 10, it is suggested to add simulation conditions such as magnetic field controls, just like the ones shown in main Fig 3.

9. The authors demonstrated a biosensing board for data collection, but in the main figures only the individual liquid sensor patch is shown. How did the authors conduct data collection, and does the liquid sensor need to be wired with the board or through wireless connection? It is suggested to add integrated images of both board and patch to show the experimental setup, either in main or supplementary figure. In addition, it is recommended to add videos of real-time data collections accordingly.

Reviewer #2 (Remarks to the Author):

In this paper, the authors introduce a liquid cardiac sensor based on permanent fluidic magnets (PFM). These PFMs, due to their properties, form a seamless interface with the wrinkles and creases on the skin surface, adapting to new folds caused by movement and featuring permanent magnetization. When subjected to minor mechanical forces, such as the pressure exerted by blood pumped to peripheral parts of the body, the alignment of dipoles within the PFM shifts, causing fluctuations in magnetic flux. These fluctuations are detected by conformal coils positioned around the sensor, generating electrical signals. These signals are recorded via a custom-designed circuit board that harnesses the magnetic flux variations generated by the PFM. The circuit includes an amplifier that magnifies the signal read from the PFM and an RC filter that removes unwanted noise. The liquid cardiac sensor offers significant advantages over traditional devices, including enhanced flexibility and adaptability to the skin, reducing artifacts from body movements and improving monitoring accuracy.

In a recent article titled "Permanent Fluidic Magnets for Liquid Bioelectronics" published in Nature Materials, the authors explained how they achieved a permanent fluidic magnet with high magnetization, flowability, and reconfigurability, and introduced its applications. In contrast, this paper focuses on applying this material to develop a cardiac monitoring device that is immune to motion artifacts. The authors use an ultrahigh-resolution Lidar scanning technique to capture tomographic images of the wrist's skin, then establish a theoretical model to understand the interaction between the reconfigurable sensor and dynamic biological tissue. They compare the signals obtained alongside a benchmark ECG, demonstrating that the liquid cardiac sensor produced stable, high-quality signals.

The presented material has many strengths in this application and holds significant potential. However, since the material and its properties have already been extensively discussed in the previous paper, this one should focus on demonstrating its applicability in the cardiac device presented. Therefore, the paper should be strengthened in aspects concerning the description and validation of the proposed device before publication. The paper is well-written, and the language is

understandable to a broad audience. However, the flow can be improved to facilitate reading, and sufficient experimental details should be provided.

Here are my detailed comments:

1. In the introduction, the authors mention the limitations of current tools but fail to mention existing non-invasive wearable devices used for ambulatory monitoring, especially those that measure the same signal they are proposing. It would be beneficial to include a discussion of these wearable competitors to clearly demonstrate the advantages of their device in comparison to the current literature.
2. At line 77 in the Results section, the phrase "In experiments, the curvature of the wrist is clearly visible in the Lidar scan data (Fig. 1e)" is reported. It is recommended to briefly clarify the experiment used to obtain this specific data.
3. In Supplementary Figure 3, from the description and the images, it is not immediately clear what exactly differentiates images (a)-(e), nor the specific variations between images (b), (d), and (e). Without further details in the text or captions, the interpretation can be ambiguous. It is recommended to explicitly specify the conditions that change between each image.
4. As previously mentioned, the description of the cardiac monitoring device needs to be improved, including details about its components. For example, in Fig. 1, a "smart device" is mentioned. What type of device and what is it used for specifically? These details should be explicitly specified to enhance clarity.
5. The board used to obtain the actual signals of interest is shown partially in Fig. 1i, Supplementary Fig. 15, and Fig. 4d. Since the authors state that it was custom-designed, more attention should be given to illustrating and explaining the choice of components. It would be beneficial to include a schematic similar to Fig. 4d, but with all components, dimensions and materials specified, along with the circuit and an explanation of how it processes the signals to produce a high-quality output.
6. The authors propose the cardiac device for ambulatory monitoring due to its extreme conformability and the seamless adaptation of the coils to the skin. However, it would be helpful to address how the rigid board necessary for signal recording is fixed in place during ambulatory monitoring. This aspect is important as it might affect the overall convenience compared to traditional rigid and bulky devices, when considering the entire system required for signal recording. Additionally, including a photo of the entire setup used to collect the signals presented in Fig. 4 and Supplementary Fig. 17 would greatly enhance clarity.
7. The type of device used for the reference ECG and how the devices were synchronized is not mentioned, so further explanation of the experimental setup is needed. Additionally, consider comparing the device with a gold standard pulse wave monitoring device, which would provide a more relevant assessment of movement artifacts.
8. Figure 5 is intended to show the relationship between time and droplet velocity, but there is no information on the specific time points from the start of the simulation to which the simulation

snapshots in panels a to h correspond. Including this information would enhance the clarity and interpretation of the figure.

9. Please specify the parameters used in the Lidar scanning, such as resolution, scanning speed, and any preprocessing steps.

10. The signals in Figure 4 should also be shown in a zoomed-in view to better appreciate the quality of the signals.

11. The authors mention "we conducted biomonitoring experiments on human subjects in various states, including sitting, standing, and walking" at lines 202 and 203, but how many subjects were involved and how many trials were conducted?

Reviewer #3 (Remarks to the Author):

The manuscript of X. Zhou et al., reports on an interesting sensor that is made of magnetic liquid. While the intent is noteworthy and the application seems unique, the manuscript seems to have significant impact on the materials science of the liquid magnets; the bioelectronic application is overstated, and the "ambulatory cardiac monitoring" device does not represent very significant results. Furthermore, on the deeper read, there is significant overlap with the previous study of the authors.

To get right into the major points:

- Three out of four figures explain the magnetic properties of the PFM liquid. Only 1.5 page describes their use for electrophysiology. Electrophysiology section also seem rather overstated – it only measures pulse wave.
- The authors do start with explanation ho the PFM liquid conforms to the non-linear surface of the skin, which would be great if the liquid was actually used on bare skin in their applications. The actual application as shown in figure 4 – is that the liquid is placed on top of another (thick, seems to be at least few 10-100s of um) of a film placed on the skin. Furthermore, that sticker has a conductive coil within, which can be used for pulse wave monitoring on itself – it seems that adding a drop of magnetizable liquid is just making the whole case very complicated to prove a case that does not need to be proven...
- To elaborate on the electrophysiological monitoring, that authors claim that "When analyzing the data, the liquid cardiac sensor even exhibited less noise than the ECG" – this is a harmful and unsubstantiated statement because authors compare the pulse wave data to ECG data. Authors do not measure ECG; hence the comparison is wrong. It is not clear how did the authors achieve the 22dB SNR – how was the data processed/analyzed?

- Furthermore, it seems like some figures and data are actually similar and slight modifications of the authors another paper [ref 23]. Just for an example, on a very quick glance, Fig. 1h = Fig. 1i (+additional dataset), and the magnetization and formation of PFM particles (Fig 2 in both manuscripts) – seems like authors just used neighboring microscopy images to explain the same effect. If the PFM effect has already been well established and presented in another work, the first 3 Figures of this manuscript could at best be condensed to a single figure to highlight the liquid PFM magnets and their major properties. Same with the manuscript itself.

Response to Referees' Comments (manuscript NCOMMS-24-26937A-Z)

We sincerely thank the reviewers and editors for their precious time and attention to our manuscript. We greatly appreciate their positive and constructive comments as well as the valuable critiques, which greatly helped us make extensive investigations in detail to elaborate our points and strengthen the manuscript. Following the reviewers' kind suggestions and guidance, we have performed additional experiments and made significant revisions to the previous manuscript. These revisions have helped consolidate our work and provide future readers with a better understanding of the liquid cardiac sensor. In the meantime, we welcome any additional concerns or suggestions regarding our revised manuscript, and we will be more than happy to address them accordingly with our continued efforts. Thank you all very much for your great contributions!

Reviewer #1 (Remarks to the Author):

The authors presented a liquid sensor for cardiac monitoring to adapt with the interfaces of biological tissues. This clever design is supported by several proof-of-concept experiments. Overall this is a well-organized and well-written manuscript with a compelling idea and well-designed supporting figures. However, there were a few major points of concern that I suggest are corrected before this paper is accepted for publication.

Response:

We sincerely thank the reviewer for his/her careful and responsible attitude toward our manuscript. Their constructive feedback and thought-provoking questions have guided us to perform extensive investigations. Additionally, we feel grateful for the reviewer's generous comments such as "*This clever design is supported by several proof-of-concept experiments*" and "*this is a well-organized and well-written manuscript with a compelling idea and well-designed supporting figures*".

1. The authors claimed that the cardiac sensor can withstand against interference such as respiration. However, it is well known that high-resolution cardiac waveforms naturally contain information of respiration, and several techniques have been developed to measure respiration rate from cardiac waveforms. It is therefore significant to check the results in this study, and modify the claims in both abstract and introduction accordingly.

Response:

We highly appreciate the reviewer for raising this insightful and meaningful question. We agree that extracting respiratory signals from cardiac waveforms is a potential alternative for measuring respiratory rate due to the physiological interaction between the cardiovascular and respiratory systems. Respiration causes the baseline of ECG signals to wander and undulate, which can be used to determine the respiration rate. Therefore, we have modified all claims related to interference against respiration accordingly.

In order to address your kind concern and benefit future readers, we have revised the Main Text.

In the Main Text, we revised the abstract as below:

“Here, we developed a reconfigurable liquid cardiac sensor capable of adapting to dynamic biological tissues, facilitating ambulatory cardiac monitoring unhindered by motion artifacts or interference from other biological activities.”

“The ECG signal consistently experienced interference as indicated by the fluctuations of the ECG signal (Supplementary Fig. 19).”

2. The monitoring of the cardiac sensor is achieved through magnetic field, but when the patch is stretched and twisted with skin during motion, the magnetic field generated will drift accordingly. It is recommended to add dynamic magnetic field distribution measurements during motion.

Response:

We highly appreciate the reviewer for raising this insightful and meaningful question. We agree with the reviewer that the magnetic field will change according to external stretching. To demonstrate this, we measured the magnetic field distribution when the sensor was stretched on the skin during motion. In our experiments, we placed the liquid sensor on the artificial skin and stretched it by 10%. The magnetic mapping of the liquid sensor was measured before and after stretching. First, we measured the magnetic mapping of the liquid sensor along its z-axis (**Fig. R1a-b**). The results showed a decrease in the sensor's magnetic field, with the profile increasing by 3%. We also measured the magnetic field strength (**Fig. R1c**), which decreased from 6.05 mT to 5.54 mT, accounting for 91.5% of its initial magnetic field. The magnetic field mapping along the y-axis was also measured (**Fig. R1d-e**), showing a similar decrease (**Fig. R1f**). In the x-axis, the decrease was less compared to the other two directions (**Fig. R1g-i**). This might be due to the stretching being applied on the y-axis. As a result, stretching the liquid sensor will slightly decrease its magnetic field, but overall, it will not compromise its performance as it can still maintain 91.5% of its initial magnetic field.

Fig. R1. Magnetic mapping of the liquid sensor. **a-b**, Magnetic mapping of the liquid sensor along its z-axis. **c**, Magnetic field strength of the liquid sensor along its z-axis before and after stretching. **d-e**, Magnetic mapping of the liquid sensor along its y-axis. **f**, Magnetic field strength of the liquid sensor along its y-axis before and after stretching. **g-h**, Magnetic mapping of the liquid sensor along its x-axis. **i**, Magnetic field strength of the liquid sensor along its x-axis before and after stretching.

In order to address your kind concern and benefit future readers, we have revised the Main Text and added Fig. R1 as Fig. 4 in the Main Text.

In the Main Text, we added a sentence below:

“Apart from that, we also investigate how the magnetic field changes in response to external stretching. To demonstrate this, we measured the magnetic field distribution when the sensor was stretched on the skin during motion. In our experiments, we placed the liquid sensor on the artificial skin and stretched it by 10%. The magnetic mapping of the liquid sensor was measured before and after stretching. First, we measured the magnetic

mapping of the liquid sensor along its z-axis (**Fig. 4a-b**). The results showed a decrease in the sensor's magnetic field, with the profile increasing by 3%. We also measured the magnetic field strength (**Fig. 4c**), which decreased from 6.05 mT to 5.54 mT, accounting for 91.5% of its initial magnetic field. The magnetic field mapping along the y-axis was also measured (**Fig. 4d-e**), showing a similar decrease (**Fig. 4f**). In the x-axis, the decrease was less compared to the other two directions (**Fig. 4g-i**). This might be due to the stretching being applied on the y-axis. As a result, stretching the liquid sensor will slightly decrease its magnetic field, but overall, it will not compromise its performance as it can still maintain 91.5% of its initial magnetic field.”

In the caption, we added a sentence below:

“**Fig. 4** | Magnetic mapping of the liquid sensor. **a-b**, Magnetic mapping of the liquid sensor along its z-axis. **c**, Magnetic field strength of the liquid sensor along its z-axis before and after stretching. **d-e**, Magnetic mapping of the liquid sensor along its y-axis. **f**, Magnetic field strength of the liquid sensor along its y-axis before and after stretching. **g-h**, Magnetic mapping of the liquid sensor along its x-axis. **i**, Magnetic field strength of the liquid sensor along its x-axis before and after stretching.”

3. In Fig 3h-j, the authors conducted finite element simulation of the liquid when applied to the wrist. How long does it take to evolve from h to j? In addition, what is the experimental surface tension of the liquid? When a liquid is dispersed onto skin due to surface tension, it is natural for the liquid to disperse differently each time during placement. Have the authors examined the reproducibility of the shape of such liquid sensor both in simulation and experiments?

Response:

We highly appreciate the reviewer for raising this insightful and meaningful question. We have added a timestamp to the simulation, indicating that it takes 1.58 seconds to evolve from **Fig. 3h** to **Fig. 3j**. The experimental surface tension of the liquid is 0.079 N/m.

We agree with the reviewer that the liquid disperses differently each time during placement. To investigate the reproducibility of the shape of the liquid sensor, **we conducted simulations** to mimic the behavior of liquid sensors dropped at different locations on the skin, observing the variation in dispersion each time. In **Fig. 3h- 3j**, the initial position was set at 10 mm. Then, we examined the reproducibility by running simulations with different initial positions of the liquid sensor, ranging from $x = 10 \text{ mm} \pm 5 \text{ mm}$. The simulation results are shown in **Fig. R2**. When the sensor was placed at positions of 15 mm, 14 mm, 6 mm, and 5 mm, it indeed formed different shapes compared to the initial position (**Fig. R2a-l**). However, despite these variations, the final formed shapes were very similar to each other (**Fig. R2c, f, i, l**).

In the experiment, we placed liquid sensors with different volumes ranging from 30 μl , 40 μl , and 50 μl of PFM at one position (**Fig. R3a-c**). We also placed the liquid sensors at different positions (**Fig. R3d-f**), and the results

showed that they formed similar shapes. In conclusion, we examined the dispersion of the liquid sensor in both experiments and simulations. Even though the liquid dispersed differently each time during placement, the final formed shapes were similar to each other.

Fig. R2. Finite element simulation of the liquid cardiac sensor. **a-c**, Finite element simulation of the liquid cardiac sensor applied to the wrist with the initial position set at 15 mm. **d-f**, Finite element simulation of the liquid cardiac sensor applied to the wrist with the initial position set at 14 mm. **g-i**, Finite element simulation of the liquid cardiac sensor applied to the wrist with the initial position set at 6 mm. **j-l**, Finite element simulation of the liquid cardiac sensor applied to the wrist with the initial position set at 5 mm.

Fig. R3. Picture of the liquid sensors dispersed on the wrist. The liquid sensors with different volumes ranging from **a**, 30 μl , **b**, 40 μl , and **c**, 50 μl of PFM at one position. The liquid sensors with different locations ranging from **d**, 0 mm, **e**, 1 mm, and **f**, 2 mm. Scale bars, 5 mm.

In order to address your kind concern and benefit future readers, we have revised both the Main Text and Supplementary Information.

In the Main Text, we added a sentence below:

“The reproducibility of the shape of the liquid sensor was investigated during each injection (Supplementary Note 3). Despite variations in different places, the final formed shapes were very similar to each other.”

In the Supplementary Information, we added a sentence below:

“Supplementary Note 3. Finite element analysis of the reproducibility of the liquid sensor.

As the liquid disperses differently each time during placement, here we investigate the reproducibility of the shape of the liquid sensor. We conducted simulations to mimic the behavior of liquid sensors dropped at different locations on the skin, observing the variation in dispersion each time. In finite element analysis, the initial position was set at 10 mm. Then, we examined the reproducibility by running simulations with different initial positions of the liquid sensor, ranging from $x = 10 \text{ mm} \pm 5 \text{ mm}$. The simulation results are shown in Supplementary Figure 23. When the sensor was placed at positions of 15 mm, 14 mm, 6 mm, and 5 mm, it indeed formed different shapes compared to the initial position (Supplementary Figure 26a-l). However, despite these variations, the final formed shapes were very similar to each other (Supplementary Figure 26c, f, i, l).”

“Supplementary Figure 26. Finite element simulation of the liquid cardiac sensor. a-c, Finite element simulation of the liquid cardiac sensor applied to the wrist with the initial position set at 15 mm. d-f, Finite element simulation of the liquid cardiac sensor applied to the wrist with the initial position set at 14 mm. g-i, Finite

element simulation of the liquid cardiac sensor applied to the wrist with the initial position set at 6 mm. j-l, Finite element simulation of the liquid cardiac sensor applied to the wrist with the initial position set at 5 mm.”

4. When the subject experiences perspiration, will sweat go into the liquid cardiac sensor and influence the measurement accuracy? In addition, the humidity of skin when placing the sensor matters – it is recommended to conduct comparative experiments on both dry and wet skin and examine the measuring accuracy.

Response:

We highly appreciate the reviewer for raising this insightful question about the influence of sweat on the liquid sensor. We have conducted experiments by applying the liquid sensors to both dry and wet skin and performing comparative tests to examine their accuracy in both scenarios. From the experiment results, we observed that the liquid sensor can attach to wet skin conformally without falling apart (**Fig. R4a-b**). This adhesion might be due to the formation of hydrogen bonds with the skin, facilitated by the water on the wet surface. Consequently, the sensor can maintain firm contact with the wet skin even when flipped over (**Fig. R4c**).

Moreover, we also examined the accuracy of the liquid sensor when sprayed with artificial sweat (**Fig. R5a**). We measured its magnetic field during the drying process (**Fig. R5b**). Initially, we observed slight fluctuations, but the magnetic field stabilized within the following three hours, demonstrating that the sensor can operate continuously without interference from sweat (**Fig. R5c**). Additionally, we also measured the generated signals on both dry and wet skin, and in both scenarios, the sensor delivered stable signals (**Fig. R5d**). As a result, the presence of sweat did not show a significant influence on the test accuracy.

Here is the process for the preparation of artificial perspiration. Artificial perspiration was used to test the stability of the liquid sensor. To prepare it, 4.65 g NaCl (Sigma Aldrich), 3.87 g 1 M lactic acid solution (Alfa Aesar), 1.80 g Urea (Alfa Aesar), 1.37 g KCl (Sigma Aldrich), 0.756 g NaHCO₃ (Sigma Aldrich), 0.546 g 1 M NH₃·H₂O (Sigma Aldrich), 0.175 g Na₂SO₄ (Sigma Aldrich), and 0.0276 g uric acid (Alfa Aesar) were added in 3 L deionized water and mixed for 30 minutes.

Fig. R4. Picture of the liquid sensor attached to wet skin. a, Wet skin. b, Liquid sensor on wet skin. Scale bars, 8 mm. c, Liquid sensor was flipped over. Scale bars, 5 mm.

Fig. R5. Stability of the liquid sensor against sweat. **a**, Picture of the PFM sprayed with artificial sweat. **b**, Picture of the PFM. Scale bars, 4 mm. **c**, Magnetic field of the PFM measured for three hours. **d**, Liquid sensor tested on dry skin and sweaty skin.

In order to address your kind concern and benefit future readers, we have revised the Main Text.

In the Main Text, we added a sentence below:

“Liquid cardiac sensor was applied on both dry and wet skin to perform comparative tests to examine their accuracy in both scenarios. The sensor can maintain firm contact with the wet skin even when flipped over (Fig. 4j). After spraying with artificial sweat, we measured its magnetic field during the drying process. Initially, we observed slight fluctuations, but the magnetic field stabilized within the following three hours, demonstrating that the sensor can operate continuously without interference from sweat (Fig. 4k). Additionally, we also measured the generated signals on both dry and wet skin, and in both scenarios, the sensor delivered stable signals (Fig. 4l). As a result, the presence of sweat did not show a significant influence on the test accuracy.”

“j, Liquid sensor was flipped over when it was attached to wet skin. Scale bars, 5 mm. k, Magnetic field of the PFM measured for three hours. l, Liquid sensor tested on dry skin and on sweaty skin.”

“Artificial perspiration. Artificial perspiration was used to test the stability of the liquid sensor. The preparation involved adding 4.65 g NaCl (Sigma Aldrich), 3.87 g 1 M lactic acid solution (Alfa Aesar), 1.80 g urea (Alfa Aesar), 1.37 g KCl (Sigma Aldrich), 0.756 g NaHCO₃ (Sigma Aldrich), 0.546 g 1 M NH₃·H₂O (Sigma Aldrich), 0.175 g Na₂SO₄ (Sigma Aldrich), and 0.0276 g uric acid (Alfa Aesar) to 3 L of deionized water and mixing for 30 minutes.”

5. What is the sampling frequency of the liquid sensor shown in Fig 4? Will the power consumption and Bluetooth transmission become a challenge when measuring such waveform continuously? It is suggested to add the lifetime of continuous measurements in the main text.

Response:

We highly appreciate the reviewer's professional questions about the sampling frequency of the liquid sensor. The sampling frequency of the liquid sensor is 1,000 Hz, and the sampling rate for the ECG is 100 Hz. The power consumption for the wireless modules is high during the pairing stage, tested to be ~19.8 mA. After the pairing stage, when measuring the waveform continuously, the current for the whole device is tested to be ~6.87 mA. Considering the battery has an output voltage of 3.7 V and a capacity of 380 mAh, we calculate the battery life given its capacity and the output current using the following formula:

$$t = \frac{C}{I} \times (1 - \eta)$$

where t is the battery life, C is the capacity of the battery, I is the output current, η is the derating factor which is set to be 10%. After calculation, the expected time for the liquid sensor is approximately 55.31 hours. Accounting for real-world inefficiencies, the battery is expected to last approximately 49.78 hours. Apart from that, wireless transmission was also not a challenge, as we set the baud rate to 115200 for the wireless modules, which meets the requirements of this device.

As a result, the battery lifetime and wireless transmission will not be a significant challenge for the device, as it can continuously operate for more than two days. To address the reviewer's question, we have added a description of the device's lifetime in the main text.

In order to address your kind concern and benefit future readers, we have revised both the Main Text and Supplementary Information.

In the Main Text, we added a sentence below:

*“Then, in order to measure the subtle pulse waves, a biosensing board was applied to connect to the conformal coil. The diagram of the biosensing board was illustrated in **Fig. 4d** and Supplementary Fig. 15. This board can work for ~55.31 hours (Supplementary Note 4).”*

In the Supplementary Information, we added Supplementary Note below:

“Supplementary Note 4. Power consumption of the biosensing board.

The power consumption for the wireless modules is high during the pairing stage, tested to be ~19.8 mA. After the pairing stage, when measuring the waveform continuously, the current for the whole device is tested to be ~6.87 mA. Considering the battery has an output voltage of 3.7 V and a capacity of 380 mAh, we calculate the battery life given its capacity and the output current using the following formula:

$$t = \frac{C}{I} \times (1 - \eta)$$

where t is the battery life, C is the capacity of the battery, I is the output current, η is the derating factor which is set to be 10%. After calculation, the expected time for the liquid sensor is approximately 55.31 hours. Accounting for real-world inefficiencies, the battery is expected to last approximately 49.78 hours. ”

6. In supplementary fig 2, it is suggested to add labels for atoms and the dimensions of the structure.

Response:

We highly appreciate the reviewer for raising this insightful and meaningful question. We have added labels for the atoms and the dimensions of the structure in the updated figures.

Fig. R6. 3D diagram of ORM network. 3D structure of ORM nanostructure formed in a carrier fluid.

Accordingly, we have updated Supplementary Figure 2 in Supplementary Information.

7. In supplementary fig 4, it is suggested to add the magnified area label of b from a. Upon comparing a (which are two spheres) and the zoomed b (which looks like irregular squares), the shape and morphology don't match each other.

Response:

We appreciate the reviewer's question. We have added a magnified area label (white square) to **Fig. R7a** from **Fig. R7b**. The original shape morphology in Supplementary Figure 4b does not match that in Supplementary Figure 4a because they were taken in different areas by Transmission Electron Microscopy (TEM). Consequently, we have put the original Supplementary Figure 4b to Supplementary Figure 4c, as it was taken at a higher magnification.

Fig. R7. TEM image of magnetic nanoparticles. a, Scale bar, 100 nm. **b**, Magnified image of the white box in 4a. Scale bar, 10 nm. **c**, Scale bar, 20 nm.

Accordingly, we have updated Supplementary Figure 4 in Supplementary Information.

8. In supplementary fig 10, it is suggested to add simulation conditions such as magnetic field controls, just like the ones shown in main Fig 3.

Response:

We highly appreciate the reviewer for his suggestion of adding simulation conditions for supplementary Figure 10. We performed a 3-dimensional Monte Carlo simulation to help study and understand the formation of the ORM network in the PFM liquid biosensors. To enhance the simulation efficiency, we have used nanomagnet clusters as the single element in the Monte Carlo simulation. A total of 500 clusters were employed in the study. All the clusters are considered as dipolar hard spheres with a radius of 1.5 μm and the system's total energy in terms of dipole-dipole interaction and truncated Lennard-Jones potential can be expressed as below,

$$U = 4\varepsilon \sum_{i,j, i \neq j} \left\{ \left(\frac{2R}{r_{ij}} \right)^{12} - \left(\frac{2R}{r_{ij}} \right)^6 - \left(\frac{2R}{r_c} \right)^{12} + \left(\frac{2R}{r_c} \right)^6 \right\} - \frac{2}{3} \pi R^3 \mu_0 \sum_i \left\{ \frac{1}{\chi} M_i^2 + R^3 \sum_{j, j \neq i} \left[\frac{(\vec{M}_i \cdot \vec{r}_{ij})(\vec{M}_j \cdot \vec{r}_{ij})}{r_{ij}^5} - \frac{\vec{M}_i \cdot \vec{M}_j}{3r_{ij}^3} \right] \right\}$$

where ε is the depth of the potential well for the truncated Lennard-Jones potential, R is the radius of the cluster, \vec{r}_{ij} is the distance between cluster i and cluster j , r_c is the cut-off radius for the cluster and equals to $2^{7/6}R$, μ_0 is the vacuum permeability. χ is the susceptibility of nanomagnets. \vec{M}_i and \vec{M}_j are the magnetizations of the nanomagnet clusters i and j . The simulation evolves the system from the totally random initial positions with the magnetization amplitude of the nanomagnet clusters assigned to be 10^7 A m^{-1} at the positive Z direction. During each simulation step, a random cluster will be selected and assigned with a small translational movement of -1.5, 0, and 1.5 μm in X, Y, and Z directions. Additionally, the magnetization of the cluster will be assigned with a random small rotation of -2, 0, and 2 degrees along the X, Y, and Z axes at each step. Each Monte-Carlo step will

be accepted if it results in a lower total energy or satisfies the acceptance probability given by $\exp(-\Delta E/k_B T)$. A total of 20,000 steps were performed to allow the system to reach equilibrium.

It is worth mentioning that the value of χ did not influence the whole simulation process because the associated item is a constant regardless of the relative positions of the nanomagnet clusters. In the simulation, we have assigned χ to be equal to 1. For the depth of the potential well ε , we assigned it to be 4.11×10^{-12} J. This value is based on the consideration that the dipole-dipole interaction of each cluster scales with the cube of the cluster radius. For ferrofluid, the typical value of ε is at the level of kT , which is 4.11×10^{-21} . Since we simulated cluster at the microscale while the ferrofluid is at the nanoscale, ε needs to be amplified by a scaling factor of 10^3 , which is the cube of the length scale. To validate the selection of the ε , we have performed a simplified 2D simulation using different values of ε : 4.11×10^{-21} , 4.11×10^{-12} , and 4.11×10^{-3} . The corresponding results are depicted in **Fig. R8**. It is clear that when ε is assigned to 4.11×10^{-12} , the simulation obtained a reasonable result. In contrast, when the ε is assigned to be kT , the nanomagnet cluster crowded together with the interparticle distance significantly smaller than the diameter. This is due to the dominating role of dipole-dipole interaction and is unrealistic. Similarly, when the ε is assigned to be 4.11×10^{-3} , the nanomagnet clusters tend to form larger clusters instead of chains. This is because of the dominant role of Lennard-Jones potential which is isotropic and is also unrealistic. One noteworthy observation in the simulation is that the formed chain is not linear, and the magnetization direction within the chain aligns with the chain's orientation. This behavior contrasts with that observed in ferrofluid simulations, thereby validating the accuracy of our simulation model.

Figure R8. 2D Monte Carlo simulation results using different ε . **a**, Results of ε using 4.11×10^{-12} . **b**, Zoomed-in view of **a**. **c**, Results of ε using 4.11×10^{-21} . **d**, Zoomed-in view of **c**. **e**, Results of ε using 4.11×10^{-3} . **f**, Zoomed-in view of **e**.

Following the reviewer's suggestion, we added the description of simulation conditions as Supplementary Note 1 in the supplementary information. We also revised the Main Text.

In the Main Text, we added a sentence below:

“We also utilized Monte Carlo simulation to replicate the formation of the network structure (Supplementary Note 1).”

9. The authors demonstrated a biosensing board for data collection, but in the main figures only the individual liquid sensor patch is shown. How did the authors conduct data collection, and does the liquid sensor need to be wired with the board or through wireless connection? It is suggested to add integrated images of both board and patch to show the experimental setup, either in main or supplementary figure. In addition, it is recommended to add videos of real-time data collections accordingly.

Response:

We greatly appreciate the reviewer for raising this insightful and meaningful question about the data collection. The liquid sensor patch was connected to a biosensing board to record the data. The board has a wireless module that can transmit the generated cardiovascular signal to smart devices such as laptops or cellphones.

For experiment details, we used conductive silver ink to make reliable connections between the biosensing board and liquid sensor patch because conductive silver ink (Conduction Inc.) is highly conductive, with a volume resistivity of less than 2×10^{-4} ohm cm. Here are the experimental methods:

First, the insulation was stripped off the ends of two wires. Then, the two wires were laid parallel to each other. Subsequently, the conductive silver ink was thoroughly mixed and applied to the two electrodes and wires to make the connection (**Fig. R9a**). The liquid sensor patch was then placed in an oven at 100 °C overnight to allow the ink to dry thoroughly (**Fig. R9b**). After drying, the wire was firmly attached to the electrodes, enabling a firm connection between the wire and the electrodes (**Fig. R9c**).

Following the reviewer's suggestions, we added the integrated images of both the biosensing board and liquid sensor patch to show the experimental setup (**Fig. R10**). We also added real-time data collection videos as well (**Supplementary Video 1**).

Fig. R9. Pictures show the connection between two electrodes and wires. a, Before put into the oven, **b,** After put into the oven. Scale bars, 0.3 mm. **c,** Picture showing the connection between the wire and electrodes. Scale bar, 2 mm.

Fig. R10. Integrated images of the experiment setup. a, Integrated image of both biosensing board and liquid sensor patch. Scale bar, 1.5 cm. **b,** Experimental setup to measure the liquid sensor and electrocardiogram (ECG). Scale bar, 5 cm.

In order to address your kind concern and benefit future readers, we have revised both the Main Text and Supplementary Information.

In the Main Text, we added a sentence below:

“Then, in order to measure the subtle pulse waves, a biosensing board was applied to connect to the conformal coil (Supplementary Fig. 15). The diagram of the biosensing board was illustrated in Fig. 4d and Supplementary Fig. 16.”

In Supplementary Information, we have added Fig. R9 as Supplementary Figure 15.

In summary, we greatly appreciate the constructive and professional comments from the reviewer on our manuscript, which helped us to make extensive investigations in detail to elaborate our points as well as fully justify the significance of this work. The insightful suggestions and corresponding revisions greatly consolidated our work to benefit future readers with a deeper understanding. Thank you very much once again! We sincerely hope that our revisions address your kind concerns.

Reviewer #2 (Remarks to the Author):

In this paper, the authors introduce a liquid cardiac sensor based on permanent fluidic magnets (PFM). These PFMs, due to their properties, form a seamless interface with the wrinkles and creases on the skin surface, adapting to new folds caused by movement and featuring permanent magnetization. When subjected to minor mechanical forces, such as the pressure exerted by blood pumped to peripheral parts of the body, the alignment of dipoles within the PFM shifts, causing fluctuations in magnetic flux. These fluctuations are detected by conformal coils positioned around the sensor, generating electrical signals. These signals are recorded via a custom-designed circuit board that harnesses the magnetic flux variations generated by the PFM. The circuit includes an amplifier that magnifies the signal read from the PFM and an RC filter that removes unwanted noise. The liquid cardiac sensor offers significant advantages over traditional devices, including enhanced flexibility and adaptability to the skin, reducing artifacts from body movements and improving monitoring accuracy. In a recent article titled "Permanent Fluidic Magnets for Liquid Bioelectronics" published in Nature Materials, the authors explained how they achieved a permanent fluidic magnet with high magnetization, flowability, and reconfigurability, and introduced its applications. In contrast, this paper focuses on applying this material to develop a cardiac monitoring device that is immune to motion artifacts. The authors use an ultrahigh-resolution Lidar scanning technique to capture tomographic images of the wrist's skin, then establish a theoretical model to understand the interaction between the reconfigurable sensor and dynamic biological tissue. They compare the signals obtained alongside a benchmark ECG, demonstrating that the liquid cardiac sensor produced stable, high-quality signals.

The presented material has many strengths in this application and holds significant potential. However, since the material and its properties have already been extensively discussed in the previous paper, this one should focus on demonstrating its applicability in the cardiac device presented. Therefore, the paper should be strengthened in aspects concerning the description and validation of the proposed device before publication. The paper is well-written, and the language is understandable to a broad audience. However, the flow can be improved to facilitate reading, and sufficient experimental details should be provided.

Response:

We sincerely thank the reviewer for his/her careful and responsible attitude toward our manuscript. Their constructive feedback and thought-provoking questions have guided us to perform extensive investigations. Additionally, we feel grateful to the reviewer's generous comments such as "*The presented material has many strengths in this application and holds significant potential*" and "*The paper is well-written, and the language is understandable to a broad audience*".

In order to fully address the reviewer's kind concern, we worked diligently to design and perform additional experiments to refine the quality of our work. The insightful suggestions and corresponding revisions greatly

consolidated our work to benefit future readers for a deeper understanding. Following the reviewer's kind suggestion, we revised the manuscript to compare it with existing non-invasive wearable devices. We also added an explanation of all components of the device.

Here are my detailed comments:

1. In the introduction, the authors mention the limitations of current tools but fail to mention existing non-invasive wearable devices used for ambulatory monitoring, especially those that measure the same signal they are proposing. It would be beneficial to include a discussion of these wearable competitors to clearly demonstrate the advantages of their device in comparison to the current literature.

Response:

We highly appreciate the reviewer's professional questions regarding the comparison with existing non-invasive wearable devices. Measuring cardiovascular signals typically involves using wearable devices such as optical effect sensors, piezoresistive effect sensors, piezoelectric effect sensors, triboelectric effect sensors, and capacitive pressure sensors. However, these devices rely on solid materials for biosensing, which are static and cannot adapt to the dynamic nature of the epidermal surface. In contrast, the liquid sensor is a novel concept that offers advantages such as low noise, high-quality signals, and adaptability to the dynamic epidermal surface.

In order to address your kind concern and benefit future readers, we have revised the Main Text.

In the Main Text, we revised a sentence below:

“Thus, recent wearable bioelectronics such as resistive^{12,13}, piezoelectric¹⁴⁻¹⁶, triboelectric¹⁷, capacitive mechanism-based sensors¹⁸⁻²⁰, and photoplethysmography (PPG)²¹ have shown great potential in allowing for non-invasive and continuous detection of human physiological signals such as blood pressure and heart rate^{22,23}.”

2. At line 77 in the Results section, the phrase "In experiments, the curvature of the wrist is clearly visible in the Lidar scan data (Fig. 1e)" is reported. It is recommended to briefly clarify the experiment used to obtain this specific data.

Response:

We greatly appreciate the reviewer for raising this professional question. We set the maximum scanning depth to 0.3 m for several reasons. First, this depth allows us to capture the curvature of the wrist with high clarity, which is essential for our analysis. A shallower scanning depth helps in reducing noise and filtering out unnecessary information that could potentially compromise the quality of the scanned data. This ensures that the data we collect is focused and relevant to the curvature of the wrist.

The scanning device we employed has a sensor size of 35.8 mm and a focal length of 30 mm. These specifications are chosen to optimize the balance between the field of view and the level of detail captured. The focal length is selected to provide the appropriate magnification for detailed scans of the wrist.

During the reconstruction process, we set the asset resolution to 2048 by 2048 pixels. This high resolution is critical for maintaining the integrity and detail of the reconstructed images, ensuring that all the details of the wrist's curvature are accurately represented. High-resolution data is vital for precise analysis and further simulation. To answer this question, we have added the experiment details in the main text as follows:

“3D scanning to capture the curvature of the wrist. The maximum scanning depth was set to be 0.3 meters, with a resolution of 2048 by 2048 pixels. The scanning device has a sensor size of 35.8 mm and a focal length of 30 mm. The scanning speed was set at 2 Hz, and the entire scanning process took 2 minutes for a single scan. Afterward, all the raw data is processed using Adobe Substance 3D Sampler tools for model reconstruction in a sequential manner.”

3. In Supplementary Figure 3, from the description and the images, it is not immediately clear what exactly differentiates images (a)-(e), nor the specific variations between images (b), (d), and (e). Without further details in the text or captions, the interpretation can be ambiguous. It is recommended to explicitly specify the conditions that change between each image.

Response:

We highly appreciate the reviewer for raising this insightful and meaningful question about the interpretation of microscopic images. The microscopic images were taken to reveal the 3D ORM network structure. Due to the depth-of-field limitation of the camera, the 3D ORM network structure cannot be captured in a single image. Therefore, it is beneficial to include multiple images to provide a comprehensive view of the network structure. In Supplementary Fig. 3a to 3d, we changed the camera's focus. Supplementary Fig. 3b, 3c, and 3e were taken at three different positions. In Supplementary Fig. 3f, we used a different lens to provide an enlarged view of the ORM network in the liquid cardiac sensor.

In order to address your kind concern and benefit future readers, we have revised the Supplementary Information. In the Supplementary Information, we added a sentence below:

“Supplementary Figures 3a to 3d were captured with the camera set to different focus points. Supplementary Figures 3b, 3c, and 3e were taken from three distinct positions. In Supplementary Figure 3f, a different lens was used to provide an enlarged view of the ORM network in the liquid cardiac sensor.”

4. As previously mentioned, the description of the cardiac monitoring device needs to be improved, including details about its components. For example, in Fig. 1, a "smart device" is mentioned. What type of device and what is it used for specifically? These details should be explicitly specified to enhance clarity.

Response:

We highly appreciate the reviewer for raising this insightful and meaningful question about the "smart device." In our demonstration, we used a laptop equipped with a signal processing and pattern recognition program designed to detect abnormalities during the monitoring process. The program first reads the pulse wave data, then normalizes the signal, and identifies key characteristics of the pulse. If any abnormalities are detected, the system will send notifications to the users. Additionally, the obtained pulse wave signals contain rich information. It was demonstrated that PWV and SI can be accurately derived from the current signal obtained using PFM (**Fig. R11**). These findings suggest that the pulse waveforms captured by our PFM liquid sensor encapsulate subtle arterial stiffness information conveyed by the forward and reflected pulse waves.

Fig. R11. Typical pulse wave profile in one cardiac cycle obtaining. a, Typical pulse wave profile in one cardiac cycle obtaining. **b,** Measured PWV and SI.

In order to address your kind concern and benefit future readers, we have revised the Main Text.

In the Main Text, we revised a sentence below:

“After the biosensing board, a smart device such as a laptop was used to record the signals for further pre-processing and to detect the presence of abnormalities.”

5. The board used to obtain the actual signals of interest is shown partially in Fig. 1i, Supplementary Fig. 15, and Fig. 4d. Since the authors state that it was custom-designed, more attention should be given to illustrating and explaining the choice of components. It would be beneficial to include a schematic similar to Fig. 4d, but with all components, dimensions and materials specified, along with the circuit and an explanation of how it processes the signals to produce a high-quality output.

Response:

We highly appreciate the reviewer for raising this meaningful question about the components, dimensions, and materials of the board. Here, we use a schematic to illustrate this (**Fig. R12**). For the integrated signal recording circuit, first, the electrical signals from the liquid sensor patch were amplified using an analog circuit (AD620).

A voltage shifter (LM358) was connected to the amplifier to shift the voltage from 0 V to 2.5 V, and a voltage follower (LM358) was used to stabilize the voltage. Subsequently, the analog signals were converted to digital signals by the embedded 12-bit analog-to-digital converter in the microcontroller (SAM D21G18) and then transmitted to the receiver unit through a 2.4 GHz wireless module (NRF24L01). The data was recorded on a laptop using a Python program. Finally, the data was processed by MATLAB 2022b for signal analysis. The dimensions of the board are 5 cm by 4 cm.

Fig. R12. Circuit diagram of biosensing board.

In order to address your kind concern and benefit future readers, we have revised the Main Text.

In the Main Text, we revised a sentence below:

“Design the circuitry. For the integrated signal recording circuit, first, the electrical signals from the liquid sensor patch were amplified using an analog circuit (AD620). A voltage shifter (LM358) was connected to the amplifier to shift the voltage from 0 V to 2.5 V, and a voltage follower (LM358) was used to stabilize the voltage. Subsequently, the analog signals were converted to digital signals by the embedded 12-bit analog-to-digital converter in the microcontroller (SAM D21G18) and then transmitted to the receiver unit through a 2.4 GHz wireless module (NRF24L01). The data was recorded on a laptop using a Python program. Finally, the data was processed by MATLAB 2022b for signal analysis. The dimensions of the board are 5 cm by 4 cm.”

6. The authors propose the cardiac device for ambulatory monitoring due to its extreme conformability and the seamless adaptation of the coils to the skin. However, it would be helpful to address how the rigid board necessary for signal recording is fixed in place during ambulatory monitoring. This aspect is important as it might affect the overall convenience compared to traditional rigid and bulky devices, when considering the entire system required for signal recording. Additionally, including a photo of the entire setup used to collect the signals presented in Fig. 4 and Supplementary Fig. 17 would greatly enhance clarity.

Response:

We greatly appreciate the reviewer for raising this insightful and meaningful question about the biosensing board for signal recording. The biosensing board was attached to the skin using a polyurethane film (MarvellHealth), ensuring a secure attachment of the biosensing board on the skin. In addition, conductive silver ink was used to make reliable connections between the biosensing board and the liquid sensor patch (**Fig. R13a**). The liquid sensor patch was placed in an oven at 100 °C overnight to allow the ink to dry thoroughly (**Fig. R13b**). After drying, the wire was firmly attached to the electrodes, enabling a secure connection (**Fig. R13c**).

Following the reviewer's suggestions, we added the integrated images of both the biosensing board and liquid sensor patch to show the entire setup for signal collection presented in Fig. 4 and Supplementary Fig. 17 (**Fig. R14**).

Fig. R13. Pictures show the connection between two electrodes and wires. a, Before put into the oven, **b**, After put into the oven. Scale bars, 0.3 mm. **c**, Picture showing the connection between the wire and electrodes. Scale bar, 2 mm.

Fig. R14. Integrated images of the experiment setup. a, Integrated image of both biosensing board and liquid sensor patch. Scale bar, 1.5 cm. **b**, Experimental setup to measure the liquid sensor and electrocardiogram (ECG). Scale bar, 5 cm.

In order to address your kind concern and benefit future readers, we have revised both the Main Text and Supplementary Information.

In the Main Text, we revised a sentence below:

“Then, in order to measure the subtle pulse waves, a biosensing board was applied to connect to the conformal coil (Supplementary Fig. 15).”

“The biosensing board was attached to the skin using a polyurethane film (MarvellHealth).”

In the Supplementary Information, we revised a sentence below:

“Supplementary Figure 15. Picture showing the connection between the wire and electrodes. Scale bar, 2 mm.”

7. The type of device used for the reference ECG and how the devices were synchronized is not mentioned, so further explanation of the experimental setup is needed. Additionally, consider comparing the device with a gold standard pulse wave monitoring device, which would provide a more relevant assessment of movement artifacts.

Response:

We highly appreciate the reviewer for raising this insightful and meaningful question about the experimental setup. All the data acquisition boards were connected to one laptop, and a Python program was used to record all the data simultaneously. The data were synchronized using a common time axis.

Currently, measuring cardiovascular signals involves Photoplethysmography (PPG), an optical technique widely recognized as a gold standard in clinical settings. We compared our device with a gold-standard PPG pulse wave monitoring device equipped with a Surface-Mount Ambient Light Photo Sensor (APDS-9008). Our findings indicate that both our device and the PPG generated stable signals during testing (**Fig. R15a**). The signals matched well with each other, indicating that the liquid cardiac sensor performed closely to the gold standard. Both sensors performed well under movement artifacts. However, PPG was prone to be influenced by varying pressures applied to the light sensor (**Fig. R15b**).

Fig. R15. Comparison between our device and a gold-standard PPG pulse wave monitoring device.

In order to address your kind concern and benefit future readers, we have revised both the Main Text and Supplementary Information.

In the Main Text, we revised a sentence below:

“In comparison to the gold standard in clinical settings such as the PPG pulse wave monitoring device, both our device and the PPG generated stable signals and matched well during testing (Supplementary Fig. 20), indicating that the liquid cardiac sensor performed well.”

“Our device was compared with a gold-standard PPG pulse wave monitoring device equipped with a Surface-Mount Ambient Light Photo Sensor (APDS-9008).”

“All the data acquisition boards were connected to one laptop, and a Python program was used to record all the data simultaneously. The data were synchronized using a common time axis.”

In the Supplementary Information, we revised a sentence below:

“Supplementary Figure 20. Comparison between our device and a gold-standard PPG pulse wave monitoring device.”

8. Figure 5 is intended to show the relationship between time and droplet velocity, but there is no information on the specific time points from the start of the simulation to which the simulation snapshots in panels a to h correspond. Including this information would enhance the clarity and interpretation of the figure.

Response:

We highly appreciate the reviewer for raising this insightful and meaningful question about the time points of the simulation. We have added specific time points in the figures (**Fig. R16**) and captions.

Fig. R16. Finite element analysis was constructed to simulate the injection process. Results show the relationship between time and droplet velocity. **a**, t = 0.05 s, **b**, t = 0.1 s, **c**, t = 0.14 s, **d**, t = 0.2 s, **e**, t = 0.5 s, **f**, t = 0.65 s, **g**, t = 0.7 s, **h**, t = 0.95 s.

In order to address your kind concern and benefit future readers, we have revised Supplementary Figures 5 and 6 in the Supplementary Information.

9. Please specify the parameters used in the Lidar scanning, such as resolution, scanning speed, and any preprocessing steps.

Response:

We greatly appreciate the reviewer's professional question about 3D scanning process. We set the maximum scanning depth to 0.3 meters, with a resolution of 2048 by 2048 pixels. The scanning speed is set at 2 Hz, and the entire scanning process takes 2 minutes for a single scan. Afterward, all the raw data is processed using Adobe Substance 3D Sampler tools for model reconstruction in a sequential manner. To answer this question, we have added this in the experiment section.

We have added the experiment details in the main text as follows:

“3D scanning to capture the curvature of the wrist. The maximum scanning depth was set to be 0.3 meters, with a resolution of 2048 by 2048 pixels. The scanning device has a sensor size of 35.8 mm and a focal length of 30 mm. The scanning speed was set at 2 Hz, and the entire scanning process took 2 minutes for a single scan. Afterward, all the raw data is processed using Adobe Substance 3D Sampler tools for model reconstruction in a sequential manner.”

10. The signals in Figure 4 should also be shown in a zoomed-in view to better appreciate the quality of the signals.

Response:

We highly appreciate the reviewer for raising this insightful and meaningful question. We have provided a zoomed-in view of the signals in Figure 4j for clearer observation (**Fig. R17**).

Fig. R17. Voltage measurements of liquid sensor and ECG for two cycles.

To answer this question, we have added the zoomed-in view to Supplementary Figure 20.

11. The authors mention "we conducted biomonitoring experiments on human subjects in various states, including sitting, standing, and walking" at lines 202 and 203, but how many subjects were involved and how many trials were conducted?

Response:

We greatly appreciate the reviewer for raising this meaningful question about the subjects and trial of our experiment. There were three subjects participating in this experiment. Each subject was tested at least three times under each scenario. The most representative signals are shown here. All cardiovascular monitoring was performed using human subjects in compliance with all the ethical regulations under protocol (ID: 20-001882) that was approved by the Institutional Review Board (IRB) at University of California, Los Angeles. All participating subjects in the study belonged to University of California, Los Angeles with the age from 20-30 years old. Population characteristics are not relevant to the experiment or the results of this study.

In order to address your kind concern and benefit future readers, we have revised the Main Text.

In the Main Text, we added a sentence below:

“Test on huamn subjects. There were three subjects participating in this experiment. Each subject was tested at least three times under each scenario. The most representative signals are shown here. All cardiovascular monitoring was performed using human subjects in compliance with all the ethical regulations under protocol (ID: 20-001882) that was approved by the Institutional Review Board (IRB) at University of California, Los Angeles. All participating subjects in the study belonged to University of California, Los Angeles with the age from 20-30 years old. Population characteristics are not relevant to the experiment or the results of this study.”

Once again, we extend our sincere appreciation to the reviewer for the valuable input and support throughout the review process which has guided us to revise our manuscript in a meaningful manner and substantially strengthened the quality of our work. We sincerely hope that our revisions address your kind concerns.

Reviewer #3 (Remarks to the Author)

The manuscript of X. Zhou et al., reports on an interesting sensor that is made of magnetic liquid. While the intent is noteworthy and the application seems unique, the manuscript seems to have significant impact on the materials science of the liquid magnets; the bioelectronic application is overstated, and the “ambulatory cardiac monitoring” device does not represent very significant results. Furthermore, on the deeper read, there is significant overlap with the previous study of the authors.

Response:

We sincerely thank the reviewer for the valuable time and attention to our manuscript. We feel inspired by your generous comments on our manuscripts as “X. Zhou et al., reports on an interesting sensor that is made of magnetic liquid”, and “intent is noteworthy and the application seems unique, the manuscript seems to have a significant impact on the materials science of the liquid magnets”. We value your critique on the bioelectronics applications, and the significance of the ambulatory cardiac monitoring results of our PFM devices, as well as the overlap with previous studies. As a result, we have made thorough revisions on the manuscript in these three aspects. First, we have condensed the sections pertaining to the study of the fundamental magnetic properties of PFM materials. Several original figures in Figures 1, 2, and 3 have been removed. Second, we have expanded and strengthened the ambulatory cardiac monitoring section of the manuscript. Specifically, we have added an additional figure (the new Figure 4) to discuss the magnetic field profile of the PFM when attached on the wrist during dynamic motions. We also validated the robust adhesion, conformability, and performance of the PFM liquid sensor on both wet and dry skins. Moreover, we discussed the importance of pulse wave monitoring and its determinative role in predicting various cardiac diseases. Then, we systematically studied the derivable parameters such as pulse wave velocity (PWV) and stiff index (SI) of pulse waveforms obtained from the PFM liquid sensor. Our results validated that the PFM liquid sensor has the unique advantage of obtaining meaningful cardiac information for disease forecasting under ambulatory conditions. Third, we revised the manuscript to more focus on the conformality of the PFM liquid materials and their suitability for wearable applications, rather than their deformability and injectability, as detailed in the previous study. This study has significantly broadened the application scenarios of PFM materials and established a new field of wearable liquid sensors.

To get right into the major points:

Three out of four figures explain the magnetic properties of the PFM liquid. Only 1.5 page describes their use for electrophysiology. Electrophysiology section also seem rather overstated – it only measures pulse wave.

Response:

We highly appreciate the reviewer for raising valuable suggestions on the structure of the manuscript. We agree with the reviewer that the sections discussing the fundamental magnetic properties of the PFM liquid should be condensed. Additionally, we recognize the importance of expanding and strengthening the section on its

application in ambulatory cardiac monitoring. Thus, in the revised manuscript, we have condensed Figures 1, 2, and 3. Meanwhile, we have incorporated a new figure illustrating the performance of PFM materials when utilized as wearable liquid sensors, emphasizing the importance of pulse wave monitoring.

We consider pulse wave measurements and analysis as critical non-invasive bioassays for the prognosis and diagnosis of cardiovascular diseases. From pulse wave measurements, several important parameters can be extracted including the PWV, pulse waveform, aortic systolic and diastolic pressures, augmentation index (AIs), and round-trip travel time of the reflecting wave. These extracted parameters can be used to evaluate arterial elasticity and stiffness. Their correlations with cardiovascular risks have been extensively validated in clinical trials^{1,2}. Additionally, pulse wave analysis has been recognized as an important supplementary method to blood pressure measurement. Blood pressure in the brachial artery fails to reveal the adverse, atherosclerotic effects of hypertension. Also, the left ventricle is primarily influenced by the pressure in the ascending aorta rather than by the pressure in the brachial artery. Differently, pulse pressure waves measured at the radial artery can be used to evaluate the aortic pressure waveform using a general transfer function^{3,4}. Our work has proved that the PFM liquid sensor can precisely measure the radial pulse waveform in ambulatory conditions. Taking a step further, we have analyzed the obtained pulse wave signals and demonstrated that PWV and SI can be accurately derived from the current signal obtained using PFM (**Fig. R18**). These findings suggest that the pulse waveforms captured by our PFM liquid sensor encapsulate subtle arterial stiffness information conveyed by the forward and reflected pulse waves. Moreover, the PFM liquid sensor approach offers additional advantages in operation, including minimal training requirements and the elimination of the need for calibration, which is necessary in application tonometry. We believe that PFM liquid sensors will bring a transformative impact to the field of pulse wave analysis, offering promising applications in predicting cardiovascular diseases and screening cardiovascular treatments.

Fig. R18. Typical pulse wave profile in one cardiac cycle obtaining. a, Typical pulse wave profile in one cardiac cycle obtaining. **b,** Measured PWV and SI.

References

1. Mitchell, G. F., Hwang, S.-J., Vasan, R. S., Larson, M. G., Pencina, M. J., Hamburg, N. M., Vita, J. A., Levy, D. & Benjamin, E. J. Arterial stiffness and cardiovascular events. *Circulation* 121, 505-511 (2010).

2. Blacher, J., Asmar, R., Djane, S., London, G. M. & Safar, M. E. Aortic pulse wave velocity as a marker of cardiovascular risk in hypertensive patients. *Hypertension* 33, 1111-1117 (1999).
3. Davies, J. I. & Struthers, A. D. Pulse wave analysis and pulse wave velocity: A critical review of their strengths and weaknesses. *Journal of Hypertension* 21 (2003).
4. Chen, C.-H., Nevo, E., Fetcs, B., Pak, P. H., Yin, F. C. P., Maughan, W. L. & Kass, D. A. Estimation of central aortic pressure waveform by mathematical transformation of radial tonometry pressure. *Circulation* 95, 1827-1836 (1997).

Following the reviewer's suggestion, we added the description of pulse waveform analysis as Supplementary Note 5 in the supplementary information. We also revised the Main Text.

In the Main Text, we added a sentence below:

“After the biosensing board, a smart device such as a laptop was used to record the signals for further pre-processing and to detect the presence of abnormalities in the pulse wave profile (Supplementary Note 5).”

In the Supplementary Information, we added a Note below:

“Supplementary Note 5. Pulse waveform analysis.

We consider pulse wave measurements and analysis as critical non-invasive bioassays for the prognosis and diagnosis of cardiovascular diseases. From pulse wave measurements, several important parameters can be extracted including the PWV, pulse waveform, aortic systolic and diastolic pressures, augmentation index (AIs), and round-trip travel time of the reflecting wave. These extracted parameters can be used to evaluate arterial elasticity and stiffness. Their correlations with cardiovascular risks have been extensively validated in clinical trials^{1,2}. Additionally, pulse wave analysis has been recognized as an important supplementary method to blood pressure measurement. Blood pressure in the brachial artery fails to reveal the adverse, atherosclerotic effects of hypertension. Also, the left ventricle is primarily influenced by the pressure in the ascending aorta rather than by the pressure in the brachial artery. Differently, pulse pressure waves measured at the radial artery can be used to evaluate the aortic pressure waveform using a general transfer function^{3,4}. Our work has proved that the PFM liquid sensor can precisely measure the radial pulse waveform in ambulatory conditions. Taking a step further, we have analyzed the obtained pulse wave signals and demonstrated that PWV and SI can be accurately derived from the current signal obtained using PFM (Supplementary Figure 27). These findings suggest that the pulse waveforms captured by our PFM liquid sensor encapsulate subtle arterial stiffness information conveyed by the forward and reflected pulse waves. Moreover, the PFM liquid sensor approach offers additional advantages in operation, including minimal training requirements and the elimination of the need for calibration, which is necessary in application tonometry. We believe that PFM liquid sensors will bring a transformative

impact to the field of pulse wave analysis, offering promising applications in predicting cardiovascular diseases and screening cardiovascular treatments.

Supplementary Figure 27. Typical pulse wave profile in one cardiac cycle obtaining. a, Typical pulse wave profile in one cardiac cycle obtaining. b, Measured PWV and SI.”

The authors do start with explanation ho the PFM liquid conforms to the non-linear surface of the skin, which would be great if the liquid was actually used on bare skin in their applications. The actual application as shown in figure 4 – is that the liquid is placed on top of another (thick, seems to be at least few 10-100s of um) of a film placed on the skin. Furthermore, that sticker has a conductive coil within, which can be used for pulse wave monitoring on itself – it seems that adding a drop of magnetizable liquid is just making the whole case very complicated to prove a case that does not need to be proven...

Response:

We highly appreciate the reviewer for raising this critical question about the setup of the liquid sensor and the film. In our experiment procedure, we placed the liquid directly on bare skin as there was a hole in the polyurethane film manufactured with a laser (**Fig. R19**). We agree with the reviewer that a conductive coil can be used for pulse wave monitoring with delicate designs. For instance, using the internal resistance of the coil to capture the pulse wave is a potential method. However, this data might not be sensitive or of high quality due to interference from motion artifacts. Resistance-based sensors typically rely on solid materials that are static and cannot adapt to the dynamic epidermal surface, resulting in a longer response time. Additionally, they need to be firmly attached to the skin, requiring a certain amount of pressure. In contrast, using a liquid cardiac sensor for pulse wave monitoring is a new concept that has not been previously considered. This design offers several advantages over traditional methods. For example, our liquid sensor recorded pulse waves with minimal interference from motion artifacts. Liquid sensor also presents a novel approach to solving conformity issues and possess the potential for accurate pulse wave monitoring.

Fig. R19. Picture showing the hole in the polyurethane film. Scale bar, 4 mm.

In order to address your kind concern and benefit future readers, we have revised the Main Text and added the detailed setup in the Supplementary Information.

In the Main Text, we revised the text as below:

“The liquid was directly put on bare skin as there was a hole in the polyurethane film manufactured with a laser.”

In the Supplementary Information, we added the picture showing the hole in the polyurethane film in Supplementary Figure 25.

To elaborate on the electrophysiological monitoring, that authors claim that “When analyzing the data, the liquid cardiac sensor even exhibited less noise than the ECG” – this is a harmful and unsubstantiated statement because authors compare the pulse wave data to ECG data. Authors do not measure ECG; hence the comparison is wrong. It is not clear how did the authors achieve the 22dB SNR – how was the data processed/analyzed?

Response:

We highly appreciate the reviewer for raising this insightful question about the comparison between the liquid sensor and the ECG, the benchmark medical testing device. We measured the ECG and liquid sensor signals from human subjects simultaneously (**Fig. R20**). All data acquisition boards were connected to one laptop, and a Python program was used to record all the data concurrently. The data were then analyzed using a common time axis, allowing us to compare the signal.

The SNR was calculated as the ratio of the power of the signal and the power of the background noise in the units of dB based on the following equation:

$$SNR = 10 \times \log_{10} \frac{P(s)}{P(n)} \dots \dots \dots (3)$$

Where $P(s)$ is the power of the measured signal and $P(n)$ is the power of the noise.

Fig. R20. Experimental setup to measure the liquid sensor and electrocardiogram (ECG). Scale bar, 5 cm.

In order to address your kind concern and benefit future readers, we have revised the Main Text.

In the Main Text, we revised the text as below:

“All the data acquisition boards were connected to one laptop, and a Python program was used to record all the data simultaneously. The data were synchronized using a common time axis.”

Furthermore, it seems like some figures and data are actually similar and slight modifications of the authors another paper [ref 23]. Just for an example, on a very quick glance, Fig. 1h = Fig. 1i (+additional dataset), and the magnetization and formation of PFM particles (Fig 2 in both manuscripts) – seems like authors just used neighboring microscopy images to explain the same effect. If the PFM effect has already been well established and presented in another work, the first 3 Figures of this manuscript could at best be condensed to a single figure to highlight the liquid PFM magnets and their major properties. Same with the manuscript itself.

Response:

We highly appreciate the reviewer for raising this insightful and meaningful question. We agree with the reviewer that the figures look similar, so we have removed the mentioned data from the main figures. However, we want to clarify that the images are not neighboring microscopy images, as they were taken from two different, independent experiments with different samples. In this work, we investigated the hysteresis loop of PFM at different temperature variations, which has not been studied before. In response to the reviewer's questions, we have condensed the first three figures, removed redundant data, and added additional figures to highlight the unique properties of the liquid cardiac sensor from the perspective of cardiac monitoring rather than focusing on the materials aspect.

In order to address your kind concern and benefit future readers, we have revised the Main Text.

In the Main Text, we revised the text as below:

“Apart from that, we also investigate how the magnetic field changes in response to external stretching. To demonstrate this, we measured the magnetic field distribution when the sensor was stretched on the skin during motion. In our experiments, we placed the liquid sensor on the artificial skin and stretched it by 10%. The magnetic mapping of the liquid sensor was measured before and after stretching. First, we measured the magnetic mapping of the liquid sensor along its z-axis (Fig. 4a-b). The results showed a decrease in the sensor's magnetic field, with the profile increasing by 3%. We also measured the magnetic field strength (Fig. 4c), which decreased from 6.05 mT to 5.54 mT, accounting for 91.5% of its initial magnetic field. The magnetic field mapping along the y-axis was also measured (Fig. 4d-e), showing a similar decrease (Fig. 4f). In the x-axis, the decrease was less compared to the other two directions (Fig. 4g-i). This might be due to the stretching being applied on the y-axis. As a result, stretching the liquid sensor will slightly decrease its magnetic field, but overall, it will not compromise its performance as it can still maintain 91.5% of its initial magnetic field.

Liquid cardiac sensor was applied on both dry and wet skin to perform comparative tests to examine their accuracy in both scenarios. The sensor can maintain firm contact with the wet skin even when flipped over (Fig. 4j). After spraying with artificial sweat, we measured its magnetic field during the drying process. Initially, we observed slight fluctuations, but the magnetic field stabilized within the following three hours, demonstrating that the sensor can operate continuously without interference from sweat (Fig. 4k). Additionally, we also measured

the generated signals on both dry and wet skin, and in both scenarios, the sensor delivered stable signals (Fig. 4I). As a result, the presence of sweat did not show a significant influence on the test accuracy.”

In the Main Text, we revised the Figures and Captions as below:

Fig. 1 | A reconfigurable and conformal liquid cardiac sensor. a, Picture of the liquid cardiac sensor. **b**, Schematic showing the gap between solid bioelectronics and skin surface. Scale bar, 1cm. **c**, Schematic showing excellent conformation of the liquid cardiac sensor droplet to topographically complex surfaces. **d**, Schematic demonstrating excellent conformation of the liquid cardiac sensor conforms to the dynamic skin surface. **e**, Plots of the outline of the skin surface. **f**, Diagram showing the alignment of nanomagnetic particles into a wavy chain network under an external impulse magnetic field. **g**, Chemical structure diagram of carrier fluids: alginate.

Fig. 2 | Dynamic magnetic field and temperature influence on ORM nanostructure. **a**, ORM structure at different temperatures ranging from 320 K ($\approx 47^\circ\text{C}$) to 260 K ($\approx -13^\circ\text{C}$). **b**, Hysteresis loop at different temperatures. **c**, A temperature change from 320 K ($\approx 47^\circ\text{C}$) to 260 K ($\approx -13^\circ\text{C}$) in the magnified area of the gray square. **d**, Magnetic pulse generated specifically to magnetize the ORM structure in the z direction only. The Inset figure is the diagram of the magnetic pulse generation by the capacitor. **e**, Diagram of a syringe filled with PFM was placed inside a coil for material manipulation. **f**, Finite element analysis was constructed to simulate the injection process. **g**, Magnetic field of single droplet as bottom left corner on Fig. i. **h**, Magnetic field of single droplet magnetized in different direction, as shown in Fig. j. **i**, Magnetic field two droplets as bottom middle section of fig. i. **j**, Magnetic field of two droplets magnetized in another direction, as shown in Fig. k. **k**, Magnetic field of tilted droplet as bottom right corner. **l**, Magnetic field of tilted droplet magnetized in another direction.

Fig. 4 | Magnetic mapping of the liquid sensor. **a-b**, Magnetic mapping of the liquid sensor **along its z-axis**. **c**, Magnetic field strength of the liquid sensor along its z-axis before and after stretching. **d-e**, Magnetic mapping of the liquid sensor **along its y-axis**. **f**, Magnetic field strength of the liquid sensor along its y-axis before and after stretching. **g-h**, Magnetic mapping of the liquid sensor **along its x-axis**. **i**, Magnetic field strength of the liquid sensor along its x-axis before and after stretching. **j**, Liquid sensor was flipped over when it was attached to wet skin. Scale bars, 5 mm. **k**, Magnetic field of the PFM in the three hours. **l**, Liquid sensor tested on dry skin and on sweaty skin.

Overall, we greatly appreciate the constructive and professional comments from the reviewers on our manuscript, which helped us to make extensive investigations in detail to elaborate our points as well as fully justify the significance of this work. We took it as our first priority and worked very hard to address the reviewers' concerns and carefully revise the manuscript accordingly. The insightful suggestions and corresponding revision greatly consolidated our work to benefit future readers with a deeper understanding. Thank you all very much! Please feel free to let us know if you have any additional concerns about our revised manuscript. We will be more than happy to revise it accordingly.

REVIEWER COMMENTS

Reviewer #1 (Remarks to the Author):

The authors have addressed all my concerns and greatly improved their manuscript. It is therefore recommended for publication as is.

Reviewer #2 (Remarks to the Author):

The authors have done an exceptional job addressing all of my comments. I recommend publishing this work in Nature Communications.

Reviewer #3 (Remarks to the Author):

I appreciate the authors' investment in the revised manuscript, significant change to its style and form, and addressing most of the comments. Additional figures and clarification are very useful, specifically the one showing a hole in the adhesive coil. However, I am still not exactly convinced by the impact and significance of the wearable monitoring system presented in this work.

- 1) Reading through the rebuttal letter, the authors claim that the ability to measure pulse waves carries information about blood flow and blood pressure, which is correct to some extent. If that would be so easy, however, I am sure authors would have shown actual Blood Pressure (BP) monitoring, but they have not; because correlating pulse wave to blood pressure is not straightforward.
- 2) It is also not possible to extend BP values from a single location; authors would need at least an array of 2 magnetic liquids measured together at a high sampling rate. Can you show at least a proof of concept that you can measure from 2 drops placed nearby and measured simultaneously? I would be excited to see this being done actually: if multi-site simultaneous sensing is possible, it would indeed open many exciting opportunities.
- 3) Authors did mention that their method provides the data "similar" to the optical PPG sensors – perhaps a direct comparison would be great to perform to actually provide quantifiable metrics.
- 4) Most importantly, the authors fail to address the significant comment about comparing pulse waves measured via magnetic liquid and ECG. We simply can not compare apples and oranges.

One is ECG, another is pulse wave. Perhaps authors might record pulse waves in another method, and compare the SNR of the pulse wave data – excellent; but comparing the SNR of the pulse wave to ECG recording is just not acceptable. ECG recordings can be compromised by a lot of additional signals: noise in the room; noise from the magnetic equipment used to measure the magnetic field; it can actually be much higher or lower, just depending on the instrumentation that ECG is recorded with, and even on the type of wires – whether authors use twisted pair or not – or perhaps how much of the open metal areas are exposed to the environment during the recording. The current comparison statements in the manuscript such as “When analyzing the data, the liquid cardiac sensor even exhibited less noise than the ECG” can not stand.

5) In the rebuttal, the authors agreed that the coil itself (without the magnetic liquid) could be used to detect pulse waves. I suggest the authors actually perform these experiments, record the pulse wave and measure its SNR; then compare that to the SNR of the magnetic liquid drop. This would be a fair comparison.

6) The authors have shown pictures of the drops of different volumes on the skin – can you provide the vital sign monitoring data with those different drops? Since the main claim of the authors is importance of the cardiac sign monitoring – it is an important factor that must be considered is: the quality of the recording with different volumes.

7) What happens when the drop is slightly smooshed/spread? Does the recording quality remain?

8) Authors also claimed the ability to record signals during movement – kindly define “movement” – I could imagine a “walking movement” when a normal swing of an arm would actually force the magnetic drop to be off the skin. If your claim is that movement artifacts disturb ECG but not magnetic readout, the comparison must be comparable.

9) Additionally, I strongly suggest authors include all additional figures that were presented in the rebuttal letter in the manuscript itself, at least in the supplementary information. Examples of those include: R20, R15b, R3-4.

Response to Reviewer #3' Remaining Comments (manuscript NCOMMS-24-26937B)

Reviewer #1 (Remarks to the Author):

The authors have addressed all my concerns and greatly improved their manuscript. It is therefore recommended for publication as is.

Response:

We sincerely appreciate the reviewer's professional suggestions for our manuscript. We also appreciate their recognition of our efforts in the revisions and acceptance of our manuscript in Nature Communications.

Reviewer #2 (Remarks to the Author):

The authors have done an exceptional job addressing all of my comments. I recommend publishing this work in Nature Communications.

Response:

We are truly grateful for the reviewer's professional comments on our manuscript. We also appreciate their acknowledgment of our work and the acceptance of our manuscript in Nature Communications.

Reviewer #3 (Remarks to the Author):

I appreciate the authors' investment in the revised manuscript, significant change to its style and form, and addressing most of the comments. Additional figures and clarification are very useful, specifically the one showing a hole in the adhesive coil. However, I am still not exactly convinced by the impact and significance of the wearable monitoring system presented in this work.

Response:

We highly appreciate the reviewer for the appreciation of our efforts in revising the manuscript. We feel grateful for the reviewer's encouragement and insightful comments, such as: "significant change to its style and form", "addressing most of the comments", and "figures and clarification are very useful". We greatly appreciate your generous comments and professional suggestions to our manuscript.

1) Reading through the rebuttal letter, the authors claim that the ability to measure pulse waves carries information about blood flow and blood pressure, which is correct to some extent. If that would be so easy, however, I am sure authors would have shown actual Blood Pressure (BP) monitoring, but they have not; because correlating pulse wave to blood pressure is not straightforward.

Response:

We highly appreciate the reviewer for raising this insightful and meaningful question. We agree with the reviewer that correlating pulse waves to blood pressure is not straightforward. However, numerous studies have shown that pulse wave analysis can be used to estimate blood pressure with considerable accuracy. For example, by combining electrocardiography and pulse wave measurements, arterial blood pressure can be inferred from pulse transit time (PTT) and the pulse waveform¹. The PTT shows a Pearson correlation coefficient of -0.712 with systolic blood pressure (SBP), while the pulse waveform demonstrates a correlation coefficient of -0.764 with diastolic blood pressure (DBP). Additionally, cuffless blood pressure can be estimated using only pulse wave analysis through parameters such as systolic upstroke time, diastolic time, and width of 1/2 pulse amplitude². The mean differences between the estimated and the measured blood pressure can reach 0.21 mm Hg for SBP and 0.02 mm Hg for DBP. The above examples clearly dictate the significance and potential clinical usage of pulse wave monitoring. Moreover, it has been demonstrated that pulse wave analysis can be utilized to estimate arterial blood pressure through the application of a general transfer function³. In this context, pulse wave monitoring is as significant as traditional cuff-based brachial blood pressure measurement, with the additional advantage of enabling continuous monitoring. Consequently, we anticipate that liquid sensors will offer a convenient and promising approach for pulse wave analysis, thereby contributing to the diagnosis and prognosis of cardiovascular diseases.

Reference

- 1 Yoon, Y., Cho, J. H. & Yoon, G. Non-constrained blood pressure monitoring using ECG and PPG for personal healthcare. *J. Med. Syst.* **33**, 261-266 (2009).
- 2 Teng, X. F. & Zhang, Y. T. in *Proceedings of the 25th Annual International Conference of the IEEE Engineering in Medicine and Biology Society (IEEE Cat. No.03CH37439)*. 3153-3156 Vol.3154.
- 3 Hirata, K., Kawakami, M. & O'Rourke, M. F. Pulse wave analysis and pulse wave velocity a review of blood pressure interpretation 100 years after Korotkov. *Circ. J.* **70**, 1231-1239 (2006).

In order to address your kind concern and benefit future readers, we have revised the Main Text.

In the discussion section of Main Text, we added a sentence below:

“Pulse wave monitoring is as significant as traditional cuff-based brachial blood pressure measurement, with the additional advantage of enabling continuous monitoring. Consequently, we anticipate that liquid sensors will offer a convenient and promising approach for pulse wave analysis, thereby contributing to the diagnosis and prognosis of cardiovascular diseases.”

2) It is also not possible to extend BP values from a single location; authors would need at least an array of 2 magnetic liquids measured together at a high sampling rate. Can you show at least a proof of concept that you can measure from 2 drops placed nearby and measured simultaneously? I would be excited to see this being done actually: if multi-site simultaneous sensing is possible, it would indeed open many exciting opportunities.

Response:

We highly appreciate the reviewer for raising this insightful and meaningful question about measuring with an array of multi-site magnetic liquids. We conducted a two-site measurement simultaneously, with one site on the forehead and another on the wrist. By measuring the pulse at these two sites with two drops of the magnetic liquid, we observed a clear time lapse of 0.06 seconds and amplitude differences between the two locations (Fig. R1). This is due to the different distances from the ventricular ejection site to the two measurement sites. Additionally, the pulse wave is weaker in the head compared to the wrist, possibly due to vascular differences in pressure. This finding proves that the multiple PFM liquid sensors can form a sensor network and are feasible for various applications, such as evaluating pulse wave velocity and assessing arterial wall stiffness. These parameters are important and independent risk factors for cardiovascular disease prognostics.

Fig. R1. Liquid sensor measured simultaneously at two sites: one on the forehead and one on the wrist.

In order to address your kind concern and benefit future readers, we have revised the Main Text. We added Fig. R1 to the Main Text as Fig. 5h.

In the Main Text, we added a sentence below:

“To show its broad application, we also conducted a two-site measurement simultaneously, with one site on the forehead and another on the wrist. By measuring the pulse at these two sites with two drops of the magnetic liquid, we observed a clear time lapse of 0.06 seconds and amplitude differences between the two locations (Fig. 5h). This is due to the different distances from the ventricular ejection site to the two measurement sites. Additionally, the pulse wave is weaker in the head compared to the wrist, possibly due to vascular differences in pressure. This finding proves that the multiple PFM liquid sensors can form a sensor network and open possibilities for various applications, such as evaluating pulse wave velocity and assessing arterial wall stiffness.”

3) Authors did mention that their method provides the data “similar” to the optical PPG sensors – perhaps a direct comparison would be great to perform to actually provide quantifiable metrics.

Response:

We highly appreciate the reviewer for raising this insightful and meaningful question about the comparison of the liquid sensors and PPG sensors. To perform a quantifiable metrics analysis, we have compared the SNR of our device and PPG sensors (Fig. R2). The liquid sensor was tested to have an SNR of 23.1 which is slightly higher than the PPG’s SNR of 18.4.

Fig. R2. Comparison of our device and PPG sensors with respect to their SNR.

In order to address your kind concern and benefit future readers, we have revised the Main Text. We added Fig. R2 to the Main Text as Fig. 51.

In the Main Text, we added a sentence below:

“The liquid sensor was tested to have an SNR of 23.1 which is slightly higher than the PPG’s SNR of 18.4 (Fig. 51).”

4) Most importantly, the authors fail to address the significant comment about comparing pulse waves measured via magnetic liquid and ECG. We simply can not compare apples and oranges. One is ECG, another is pulse wave. Perhaps authors might record pulse waves in another method, and compare the SNR of the pulse wave data – excellent; but comparing the SNR of the pulse wave to ECG recording is just not acceptable. ECG recordings can be compromised by a lot of additional signals: noise in the room; noise from the magnetic equipment used to measure the magnetic field; it can actually be much higher or lower, just depending on the instrumentation that ECG is record with, and even on the type of wires – whether authors use twisted pair or not – or perhaps how much of the open metal areas are exposed to the environment during the recording. The current comparison

statements in the manuscript such as “When analyzing the data, the liquid cardiac sensor even exhibited less noise than the ECG” can not stand.

Response:

We highly appreciate the reviewer for raising this insightful and meaningful concern. We agree with the reviewer that ECG signals and the pulse wave measured via magnetic liquids are different physiological signals. ECG records electrical signals of the heart pumping while the PFM liquid sensor records pulse waves containing information of blood flow changes caused by the heart pumping. Although both signals are related to heart activity and reflect cardiovascular health status, their signals might not be compared directly. Therefore, we have removed the comparison of pulse wave and ECG signals in terms of their performance and revised the statement in the manuscript as below,

we revised,

“When analyzing the data, the liquid cardiac sensor even exhibited less noise than the ECG”

to

“The liquid cardiac sensor provides stable signals with minimum noise. Therefore, it can be combined with the ECG measurement to provide a comprehensive evaluation of the cardiovascular health status of human beings.”

Accordingly, we have extensively modified Figure 5 to remove the comparison of ECG signals and PFM signals.

5) In the rebuttal, the authors agreed that the coil itself (without the magnetic liquid) could be used to detect pulse waves. I suggest the authors actually perform these experiments, record the pulse wave and measure its SNR; then compare that to the SNR of the magnetic liquid drop. This would be a fair comparison.

Response:

We appreciate the reviewer for raising this insightful question. We agree that a conductive coil could potentially be used for pulse wave monitoring with careful design considerations. However, in our experimental setup, the coil consists of straight wire, and its resistance remains almost unaltered to the subtle pulse wave when firmly attached to the skin (**Fig. R3**). To measure the pulse wave using only a coil, a more complex design with different structures and materials will be necessary, rather than just the coated straight metal wire. In our experiment, the metal coil satisfies our study requirements as its resistance remains almost unaltered to pulse waves, thereby minimizing any interference. As a result, the SNR of the coil itself is ~ 0 . The SNR of the magnetic liquid drop is 23.1.

Fig. R3. Using coil for pulse wave monitoring.

In order to address your kind concern and benefit future readers, we have revised the Main Text. We added Fig. R3 to the Main Text as Fig. 5e.

In the Main Text, we added a sentence below:

“To remove the interference of the coil, we also measured the pulse wave using only a coil that consists of straight wire. It did not yield any pulse wave signals (Fig. 5e). Thus, the metal coil satisfies our study requirements as its resistance remains almost unaltered to pulse wave, thereby minimizing any interference.”

6) The authors have shown pictures of the drops of different volumes on the skin – can you provide the vital sign monitoring data with those different drops? Since the main claim of the authors is importance of the cardiac sign monitoring – it is an important factor that must be considered is: the quality of the recording with different volumes.

Response:

We highly appreciate the reviewer for raising this professional question about measuring with different volumes. We have compared the quality of the recordings with volumes of 20 μl , 30 μl , and 50 μl . It is clear that, with the increase in volume, the liquid sensor demonstrates higher electrical output and better signal quality (**Fig. R4**). This is because as the volume of PFM increases, it forms a stronger magnetic field, thereby generating higher electrical output.

Fig. R4. Testing of the liquid sensor with different PFM volumes of 20 µl, 30 µl, and 50 µl.

In order to address your kind concern and benefit future readers, we have revised the Main Text. We added Fig. R4 to the Main Text as Fig. 5g.

In the Main Text, we added a sentence below:

“We have compared the quality of the recordings with volumes of 20 µl, 30 µl, and 50 µl. With the increase in volume, the liquid sensor demonstrates higher electrical output and better signal quality (Fig. 5g). This is because as the volume of PFM increases, it forms a stronger magnetic field, thereby generating higher electrical output.”

7) What happens when the drop is slightly smooshed/spread? Does the recording quality remain?

Response:

We highly appreciate the reviewer for raising this insightful and meaningful question. To address the reviewer's concern, we conducted a systematic experiment to assess the magnetic field strength of the liquid sensor after it was spread. The experiment demonstrated that the magnetic field strength decreased when the liquid was spread (Fig. R5a). At the same time, we also found that the spreading of PFM will result in reduced performance in its electrical performance, leading to a slight decrease in recording quality (Fig. R5b). However, the recording quality remained acceptable.

Fig. R5. Testing of the liquid sensor after spreading. **a**, Magnetic field of the PFM measured when it was slightly spread. **b**, Liquid sensor tested before and after spread.

In order to address your kind concern and benefit future readers, we have revised the Main Text. We added Fig. R5a to the Main Text as Fig. 5i. We added Fig. R5b to the Main Text as Fig. 5j.

In the Main Text, we added a sentence below:

“To understand the performance of the liquid cardiac sensor after it was spread, we assessed its magnetic field strength. The experiment demonstrated that the magnetic field strength decreased when the liquid was spread (Fig. 5i). At the same time, we also found that the spreading of PFM resulted in reduced performance in its electrical signal, leading to a slight decrease in recording quality (Fig. 5j). However, the recording quality remained acceptable.”

8) Authors also claimed the ability to record signals during movement – kindly define “movement” – I could imagine a “walking movement” when a normal swing of an arm would actually force the magnetic drop to be off the skin. If your claim is that movement artifacts disturb ECG but not magnetic readout, the comparison must be comparable.

Response:

We highly appreciate the reviewer for raising this insightful and meaningful question. We have removed the claims about walking movement and the assertion that movement artifacts disturb ECG readouts from the article. The comparison between the ECG and liquid sensor has been replaced with other meaningful data in this response. To address your concern and benefit future readers, we have revised the description in the Main Text and Fig. 5 accordingly.

In the Main Text, we added a paragraph below:

“We have compared the quality of the recordings with volumes of 20 μ l, 30 μ l, and 50 μ l. With the increase in volume, the liquid sensor demonstrates higher electrical output and better signal quality (Fig. 5g). This is because as the volume of PFM increases, it forms a stronger magnetic field, thereby generating higher electrical output. To show its broad application, we also conducted a two-site measurement simultaneously, with one site on the forehead and another on the wrist. By measuring the pulse at these two sites with two drops of the magnetic liquid, we observed a clear time lapse of 0.06 seconds and amplitude differences between the two locations (Fig. 5h). This is due to the different distances from the ventricular ejection site to the two measurement sites. Additionally, the pulse wave is weaker in the head compared to the wrist, possibly due to vascular differences in pressure. This finding proves that the multiple PFM liquid sensors can form a sensor network and are feasible for various applications, such as evaluating pulse wave velocity and assessing arterial wall stiffness. These parameters are important and independent risk factors for cardiovascular disease prognostics. To understand the performance of the liquid cardiac sensor after it was spread, we assessed its magnetic field strength. The experiment demonstrated that the magnetic field strength decreased when the liquid was spread (Fig. 5i). At the same time, we also found that the spreading of PFM resulted in reduced performance in its electrical signal, leading to a slight decrease in recording quality (Fig. 5j). However, the recording quality remained acceptable..”

9) Additionally, I strongly suggest authors include all additional figures that were presented in the rebuttal letter in the manuscript itself, at least in the supplementary information. Examples of those include: R20, R15b, R3-4.

Response:

We thank the reviewer for raising this question about including additional figures in the manuscript and Supplementary Information. Following the reviewer’s question, we have moved all the figures inside the previous response (R20, R15b, and R3-4) into the supplementary information. We also moved additional figures in this response from R1 to R5 into the manuscript.

In order to address your kind concern and benefit future readers, we have revised both the Main Text and Supplementary Information.

In the Main Text, we revised the Fig. 5 below:

“Fig. 5 | PFM characterization and signal output. a, 3D scanning technique to capture tomographic images showing the skin surface. Scale bar, 8 mm. b, Enlarged view from of the skin surface. Scale bar, 3 mm. c, Image of the liquid cardiac sensor. Scale bar, 8 mm. d, Circuit diagram of liquid cardiac sensor. e, Using coil for pulse wave monitoring. f, Characteristics of pulse wave from the liquid sensor. g, Testing of the liquid sensor with different PFM volumes of 20 μl , 30 μl , and 50 μl . h, Liquid sensor measured simultaneously at two sites: one on the forehead and one on the wrist. i, Magnetic field of the PFM measured when it was slightly spread. j, Liquid sensor tested before spread and after spread. k, Comparison between our device and a gold-standard PPG pulse wave monitoring device. l, Comparison of our device and PPG sensors with respect to their SNR.”

In the Supplementary Information, we added Figures R3-4, R15b, and R20 in previous response as Supplementary Figures 14, 18, 19 and 24 as below:

“Supplementary Figure 14. Picture of the liquid sensor attached to wet skin. a, Wet skin. b, Liquid sensor on wet skin. Scale bars, 8 mm.”

“Supplementary Figure 19. Comparison between our device and a gold-standard PPG pulse wave monitoring device. a, Comparison between the liquid sensor and PPG pulse wave monitoring device. b, PPG pulse wave monitoring device under stress of 50 Pa and 10 kPa.”

“Supplementary Figure 24. Picture of the liquid sensors dispersed on the wrist. The liquid sensors with different volumes ranging from a, 30 μl, b, 40 μl, and c, 50 μl of PFM at one position. The liquid sensors with different locations ranging from d, 0 mm, e, 1 mm, and f, 2 mm. Scale bars, 5 mm.”

We have added a sentence in Supplementary Note 3 as below,

“In the experiment, we placed liquid sensors with different volumes ranging from 30 μl, 40 μl, and 50 μl of PFM at one position (Supplementary Figure 24a-c). We also placed the liquid sensors at different positions (Supplementary Figure 24d-f), and the results showed that they formed similar shapes.”

In summary, we greatly appreciate the constructive and professional comments from the Reviewer 3 on our manuscript, which helped us to make extensive investigations in detail to elaborate our points as well as fully justify the significance of this work. Thank you very much once again! We sincerely hope that our revisions address your kind concerns.

REVIEWERS' COMMENTS

Reviewer #3 (Remarks to the Author):

I very much appreciate the authors' constructive response to the comments and according changes to the manuscript. I would recommend publishing the work at this stage.